



# The Making of the New European Wind Atlas, Part 1: Model Sensitivity

Andrea N. Hahmann[1], Tija Sīle[2], Björn Witha[3,4], Neil N. Davis[1], Martin Dörenkämper[5], Yasemin Ezber[6], Elena García-Bustamante[7], J. Fidel González-Rouco[8,9], Jorge Navarro[7], Bjarke T. Olsen[1], and Stefan Söderberg[10,11]

[1]Wind Energy Department, Technical University of Denmark, Roskilde, Denmark
[2]Institute of Numerical Modelling, Department of Physics, University of Latvia, Riga, Latvia
[3]ForWind, Carl von Ossietzky University Oldenburg, Germany
[4]energy & meteo systems GmbH, Oldenburg, Germany
[5]Fraunhofer Institute for Wind Energy Systems, Oldenburg, Germany
[6]Eurasia Institute of Earth Sciences, Istanbul Technical University, Istanbul, Turkey
[7]Wind Energy Unit, CIEMAT, Madrid, Spain
[8]Dept. of Earth Physics and Astrophysics, University Complutense of Madrid, Madrid, Spain
[9]Institute of Geosciences, IGEO (UCM-CSIC), Madrid, Spain
[10]WeatherTech, Sweden
[11]Renewable Energy Analytics, DNV-GL Energy, Sweden

**Correspondence:** Andrea N. Hahmann (ahah@dtu.dk)

**Abstract.** This is the first of two papers that documents the creation of the New European Wind Atlas (NEWA). It describes the sensitivity analysis and evaluation procedures that formed the basis for choosing the final setup of the mesoscale model simulations of the wind atlas. An optimal combination of model setup and parameterisations was found for simulating the climatology of the wind field at turbine-relevant heights with the Weather Research and Forecasting (WRF) model. Initial WRF model sensitivity experiments compared the wind climate generated by using two commonly used planetary boundary layer schemes and were carried out over several regions in Europe. They confirmed that the largest differences in annual mean wind speed at 100 m above ground level mostly coincide with areas of high surface roughness length and not with the location of the domains or maximum wind speed. Then an ensemble of more than 50 simulations with different setups for a single year were carried out for one domain covering Northern Europe, for which tall mast observations were available. Many different parameters were varied across the simulations, for example model version, forcing data, various physical parameterisations and the size of the model domain. These simulations showed that although virtually every parameter change affects the results in some way, significant changes on the wind climate in the boundary layer are mostly due to using different physical parameterisations, especially the planetary boundary layer scheme, the representation of the land surface, and the prescribed surface roughness length. Also the setup of the simulations, such as the integration length and the domain size can considerably influence the results. The degree of similarity between winds simulated by the WRF ensemble members and the observations was assessed using a suite of metrics, including the Earth Mover's Distance (EMD), a statistic that measures the distance between two probability distributions. The EMD was used to diagnose the performance of each ensemble member using the full wind speed distribution, which is important for wind resource assessment. The most realistic ensemble members





were identified to determine the most suitable configuration to be used in the final production run, which is fully described and evaluated in the second part of this study (Dörenkämper et al., 2020).

## 1 Introduction

Wind atlases can be defined as databases of wind speed and direction statistics at several heights in the planetary boundary layer. These atlases have been created for many regions of the world mainly to help inform wind energy installations, but not only, many other human activities benefit from the knowledge of the wind behaviour at its climatology. For example, for structural design for buildings, transportation infrastructure and operation, recreation and tourism, to name a few. In 1989, the European Wind Atlas (EWA, Troen and Petersen, 1989) was released, which provided the first wind atlas covering all of

Europe. The EWA was mostly based on surface observations, and used the so-called Wind Atlas method (Troen and Petersen, 1989), which makes it possible to transfer detailed information about the mean wind climate from one location to another. Before the New European Wind Atlas (NEWA[a], Petersen, 2017), the EWA was the only public source of wind climate data that covered the whole of Europe in a homogeneous way. Given that the EWA is now 30 years old, it is lacking information that limits it usefulness, that is it has a very coarse spatial resolution, does not provide time series of the variables of interest, and

was developed using a linearised model, which limited its applicability in complex terrain. Nowadays, modern wind atlases rely on the output from mesoscale model simulations, either sampling recurrent atmospheric states (e.g., Frank and Landberg, 1997; Pinard et al., 2005; Badger et al., 2014; Chávez-Arroyo et al., 2015) or by long-term simulations with Numerical Weather Prediction (NWP) models (e.g., Tammelin et al., 2013; Nawri et al., 2014; Hahmann et al., 2015; Draxl et al., 2015; Wijnant et al., 2019). NEWA follows the latter approach and provides a unified high-resolution and publicly available dataset of wind

resource parameters covering all of Europe and Turkey. The wind atlas is based on 30 years of mesoscale model simulations with the Weather Research and Forecasting (WRF, Skamarock et al., 2008) model at $3\,\mathrm{km} \times 3\,\mathrm{km}$ spatial and $30\,\mathrm{min}$ temporal resolution and 7 vertical levels. Wind statistics from further downscaling with a microscale model (see Dörenkämper et al., 2020, for more details) are provided for Europe and Turkey onshore and offshore up to $100\,\mathrm{km}$ from the coastline, plus the Baltic and the North Seas with a horizontal grid spacing of $50\,\mathrm{m}$ at three wind-turbine relevant heights.

Mesoscale models are, in general, not specifically developed for wind energy applications; however, over the last decade they have been extensively used for that purpose (see Olsen et al., 2017, for a review). Developing an optimal WRF model configuration for wind resource assessment is not a straightforward task, considering the large number of degrees of freedom in the model configuration, and the different choices of input data. Among the configuration options offered in the WRF model are, physical parameterisations such as planetary boundary layer (PBL), surface layer (SL), land surface model (LSM), cloud

micro-physics, and radiation. Also numerical and technical options (e.g., domain layout, nudging options, time step), and the

---
[a]https://map.neweuropeanwindatlas.eu/





initial and boundary conditions of the atmosphere, sea surface, and land surface are relevant aspects to be explored before determining the set up that better fits a specific application. It is impossible to test every combination of these parameters, as the number of such experiments would be in the thousands, which is unfeasible in terms of computational resources. Therefore, a compromise between available computational power and scientific soundness had to be found. The approach in NEWA was

to first define a "best practice" setup using the vast and diverse experience of the mesoscale modellers in the project, and then to test the sensitivity of the results to changes in the model configuration that are presumably the most relevant for the simulation of the wind field. This includes some physical options, such as PBL schemes, but also included a wide range of other parameters, such as numerical options, for which sensitivity results are rarely reported in the literature. Those parameters that did not evidence an impact in the simulation of the wind field where fixed as in the best practice set up. It is impossible

to claim that all existing sensitivities in the model were found and tested, however the large number of parameters that were tested and found not to be influential gives some credibility to our approach.

Large number of ensembles of model simulations using the WRF model are documented in the literature in many applications. The WRF model is also used for more general climate research purposes as a regional climate model (RCM), for example, within the context of the Coordinated Regional Climate Downscaling Experiment (CORDEX) project (Katragkou

et al., 2015), however, in such cases the attention is typically focused on climate-relevant parameters, such as temperature or precipitation. A number of sensitivity studies for the dependence of model-simulated temperature and precipitation on cloud microphysics, convection and radiation schemes (Katragkou et al., 2015), or PBL schemes (García-Díez et al., 2013) and all of the above (Strobach and Bel, 2019) has been reported. These show that the biases in model results can depend on the model setup, study region, season or diurnal cycle. Additionally, in some cases, it is suspected that reduction of bias in a specific

setup can be caused by errors in two different processes having the opposite effect and cancelling each other out. There are cases where all model setups fail to replicate some aspect correctly when compared to observations, e.g. as shown in (Mooney et al., 2017) for the diurnal cycle of precipitation. Sensitivity studies with a large number of WRF model simulations for wind energy applications have also been reported in Lee et al. (2012); Siuta et al. (2017); Fernández-González et al. (2017, 2018) for PBL wind ensemble prediction. Very few studies afforded an exhaustive sensitivity analysis of the model performance

on the near-surface long-term wind climatology and many lack the verification at wind turbine heights. Two examples that looked at the sensitivity of the modelled wind climate are Hahmann et al. (2015), who investigated a limited number of model parameters over the sea in Northern Europe and Floors et al. (2018b), who concentrated on the impact of the model's spatial resolution on the coastal flow. This study expands on these earlier attempts with a much larger set of sensitivity simulations and the comparison to observations for the wide European domain.

With this background, our objective is to summarise the mesoscale simulations that form the backbone of the New European Wind Atlas. The scope of mesoscale modelling in NEWA is ambitious and cannot be addressed within one article. Therefore it is divided into two parts. The first one (this article) deals with sensitivity simulations and the second part (Dörenkämper et al., 2020) describes the production run and its evaluation.

All simulations in this study covered one full year and used similar grid parameters and modelling setup, which will be briefly

described in Section 2. The data used for the evaluation of the ensemble of simulations among the whole pool of cases that



are best suited to provide a meaningful sensitivity range is presented in Section 3; the statistics used in the model assessment and comparison among ensemble members is introduced in Section 4. The process of finding the most adequate (in a sense that will be defined) combination of model setup and parameterisations occurred in several phases: (1) analysis of sensitivity dependence on the geographical domain (Section 5.1), (2) selection of the WRF model version (Section 5.2), (3) creation of a large multi-physics ensemble (Sections 5.3 and 5.4) and (4) the analysis of the model sensitivity to the size of the model domain (Section 5.5). A summary of the findings of the sensitivity experiments can be found in Section 5.6. The paper ends with a discussion of the limitations of the approach used and the outlook (Section 6).

## 2 Description of the WRF model simulation

The database of simulated winds and wind-energy relevant parameters for the model sensitivity tests was created by splitting the simulation period into a series of relatively short WRF model runs that, after concatenation, cover at least a year. The simulations overlap in time during the spin-up period, typically between 3 to 24 h, which is discarded, as described in (Hahmann et al., 2010, 2015; Jiménez et al., 2010, 2013). Two approaches are tested: frequent re-initialisation, in our case daily 36-hour runs, versus several days long runs that are nudged towards the forcing reanalysis. In the first approach, the re-initialisation every day keeps the runs close to the driving reanalysis and the model solution is free to develop its own internal variability. In the second approach, the use of nudging prevents the model solution from drifting from the observed large-scale atmospheric patterns, and the multi-days simulation ensures that the mesoscale flow is fully in equilibrium with the mesoscale characteristics of the terrain (Vincent and Hahmann, 2015). Both methods have the added advantage that the simulations are independent of each other, and therefore, can be computed in parallel, reducing the total time needed to complete a multi-year climatology. In comparison, in continuous regional climate simulations a single run can last several wall clock months. Although these type of continuous runs might present certain advantages, as for instance that they preserve the memory of land-atmosphere processes, the selection of one approach or the other depends upon the needs of the specific experiment.

All mesoscale simulations in NEWA used three nested domains with a 3 km horizontal grid spacing for the innermost grid and a 1:3 ratio between inner and outer domain resolution, leading to 3 different resolutions: 27 km for the outer domain, and 9 km and 3 km for the inner nested domains. The model top was set to 50 HPa, following the best practices recommended by the WRF developers (Wang et al., 2019). The temporal coverage of the simulations is one year (2015). Other parameters common to all simulations are listed in Table A1. We explore the effect of changing various relevant parameters of the simulation set up from the base model configuration explained above to estimate the wind climatology over Europe.

## 3 Observed data

High-quality data from tall masts for the evaluation of the various sensitivity experiments is rare. In this study we used data from eight sites in northern Europe. The locations and names of the sites are shown in Figure 1; the details of the sites are





summarised in Table 1. All sites are equipped with towers, and IJmuiden has an additional Zephir 300 continuous-wave lidar recording wind speed and direction at 25 m intervals between 90 and 315 m (Kalverla et al., 2017).

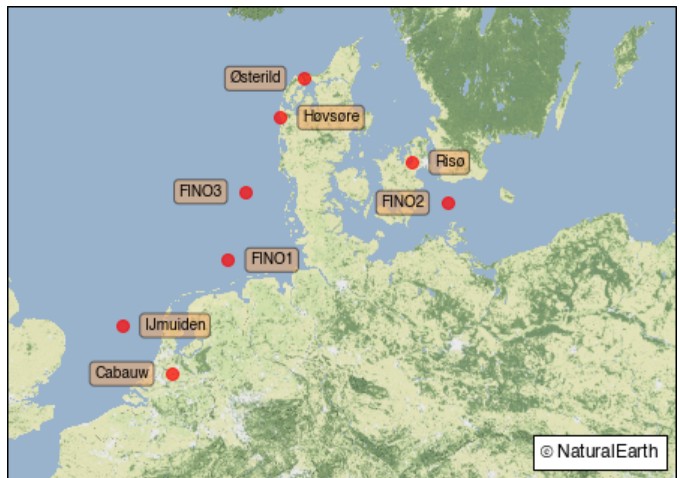

**Figure 1.** Location and name of the tall mast/lidar sites used in the model evaluation. Base map created with Natural Earth.

**Table 1.** Tall mast sites and wind speed measurement heights available at each site. The values indicated in bold are used in the evaluation of wind profiles; the values underlined are those used in the evaluation of temporal variability. For both cases the sample size is indicated in the last column.

| site | type | measurement heights (m AGL/AMSL) | sample size (profile) | sample size (time series) |
|---|---|---|---|---|
| FINO1[a] | Offshore | 34, **41.5**, **51.5**, **61.5**, **71.5**, **81.5**, **91.5**, 104.5 | 5987 | 6207 |
| FINO2[a] | Offshore | 32.4, **42.4**, **52.4**, **62.4**, **72.4**, **82.4**, **92.4**, 102.5 | 6842 | 8091 |
| FINO3[a] | Offshore | 30, 40, **50**, 60, **70**, 80, **90**, 100, 106 | 8436 | 8441 |
| IJmuiden[b] | Offshore | **27**, **58**, **89**, **115**, **140**, **165**, **190**, **215**, **240**, **265**, 290, 315 | 6952 | 8424 |
| Høvsøre[c] | Coastal | 10, **40**, **60**, **80**, **100**, 116.5 | 8147 | 8164 |
| Risø[d] | Land | **44.2**, **76.6**, **94**, **118**, **125.2** | 7779 | 7791 |
| Østerild[e] | Land | 10, **40**, **70**, **106**, **140**, **178**, **210**, **244** | 6822 | 6935 |
| Cabauw[f] | Land | 10, **20**, **40**, **80**, **140**, **200** | 8517 | 8551 |

[a]https://www.fino-offshore.de/en/, [b]Kalverla et al. (2017), [c]Peña et al. (2015),

[d]http://rodeo.dtu.dk/rodeo/ProjectOverview.aspx?&Project=5&Rnd=674271, [e]Peña (2019) [f]http://www.cesar-database.nl

High quality data is essential for model evaluation (Lucio-Eceiza et al., 2018a, b). The mast data has been quality controlled and the mast flow distortion on the wind speed estimates was minimised by sub-setting the data. At FINO1, FINO2, Risø and Høvsøre, where wind speed measurements are available from only one boom, winds originating from ± 10° of the boom direction are filtered. At FINO3, we use the data from the three heights where wind speed measurements from three booms are





available. The wind speed value is taken from the boom where the wind direction is most perpendicular to the boom direction. At IJmuiden, the data was processed as discussed in Kalverla et al. (2017). At Cabauw, the data was processed and gap filled as described in (Bosveld, 2019).

In addition to mast flow distortion, the wind speed estimates at some of the measurement sites are impacted by nearby
wind farms. At Høvsøre and Østerild, test turbines are located north of the mast, in the sector with the least frequent wind directions. At FINO1, the wind farm *Alpha Ventus* impacts the eastern sector with the nearest turbine only 405 m away in the direction of 90°. At FINO3, the *DanTysk* wind farm went into operation in 2015 to the east of the mast. Due to the difficulty in understanding the impacts of the wind turbines on the measurements, the data has not been filtered or corrected for the turbine wakes.

All measurement data had 10 min mean values. The measurement data was filtered to one period per hour, using the period that was closest to 00 minutes, giving a maximum sample size of 8760. The sample size in Table 1 represents the number of samples when all levels used for the wind profile evaluation, indicated in bold, are available. The filtering for mast flow distortion process removes a small number of samples. For the time-series evaluation (e.g. correlation and RMSE), we used measurements from the level that was closest to 100 m above mean sea level and had a good data availability (underlined
values, Table 1).

## 4 Model evaluation metrics

There are many ways of comparing two time series, and the best mathematical tool for such a comparison depends on the relationship being compared. The main goal of the NEWA project was the evaluation of the wind climate, which is usually understood as the probability distribution of wind speed and direction at a specific point. The temporal mean of each modelled
distribution, $\overline{u}_m$, and the observed distribution, $\overline{u}_o$, was calculated for identical periods. The bias herein is defined as difference between the two means, $\overline{u}_m - \overline{u}_o$. If the bias is positive, the model overestimates the observed wind speed.

In some applications the temporal accuracy could be important. Although not our primary focus, we calculate time series metrics to gauge the overall quality of the results. If the relative performance of in time-dependent metrics would be significantly different from performance climate wise, that would indicate deep problems in models and would complicate the
decisions. However, broadly speaking, that is not the case in our results. The information about temporal co-variability is provided herein by the Pearson correlation coefficient, $r$. The root mean square error (RMSE) is used to provide an estimate of systematic biases in model skill (von Storch and Zwiers, 1999). The RMSE is calculated over all $i$ time steps, with $u_o^i$ and $u_m^i$ being the $i$-th modelled and observed values in the time series of length $n$. The RMSE can be calculated as:

$$\text{RMSE} = \sqrt{\frac{1}{n}\sum_{i=1}^{n}(u_m^i - u_o^i)^2}. \tag{1}$$

In the context of this study, a large ensemble of WRF model setups will be compared against the observations at a number of sites. As stated above, one of the ensemble members was designated to be the baseline or "base". The aim is therefore to evaluate if a certain model set up, from the pool, performs better than the baseline configuration. This can be described for a





generalised error statistic $m$ defined for the baseline as $m_B$ and $m_{M_j}$ for the $j$-th ensemble member. If the statistic $m$ has the property that the ideal result is $m = 0$ and $m > 0$ means degraded results with respect to the baseline case, then for negative values of the ratio $(m_{M_j} - m_B)/m_B$, the $j$-th member performs better than the baseline and the opposite is also true. Thus, the value of this ratio is the relative improvement or worsening, in percent, compared against the baseline.

5    When assessing the change in bias between the $j$-th ensemble member and the baseline, we use the difference in the absolute relative bias, $|(\overline{u}_j - \overline{u}_o)/\overline{u}_o| - |(\overline{u}_B - \overline{u}_o)/\overline{u}_o|$, where the overbar denotes the temporal mean. If the relative bias of the $j$-th ensemble is closer to zero than that of the base, then the $j$-th ensemble member is closer to the observations than the baseline case and the difference of the absolute biases will be negative.

While the bias is a popular error statistic for comparing two distributions of wind speed, it suffers from the fact that two distributions can have the same means while having completely different shapes (see Fig. 2 and discussion below). The shape of the wind speed distribution plays a large role in wind energy applications, since wind power is proportional to the cube of the wind speed; this results in small changes in the wind speed distribution being amplified when converted to power.

Therefore, we also applied the Earth Mover's distance (EMD, Rubner et al., 2000) metric, which is popular in image processing, to evaluate the shape of the distribution. This metric can be understood as the amount of physical work needed to move a pile of soil in the shape of one distribution to that of another distribution. The EMD is equivalent to the area between two cumulative distribution functions, and, with slight modifications, can be applied to circular variables (Rabin et al., 2008). The EMD calculation was calculated using the Pyemd package (Pele and Werman, 2008).

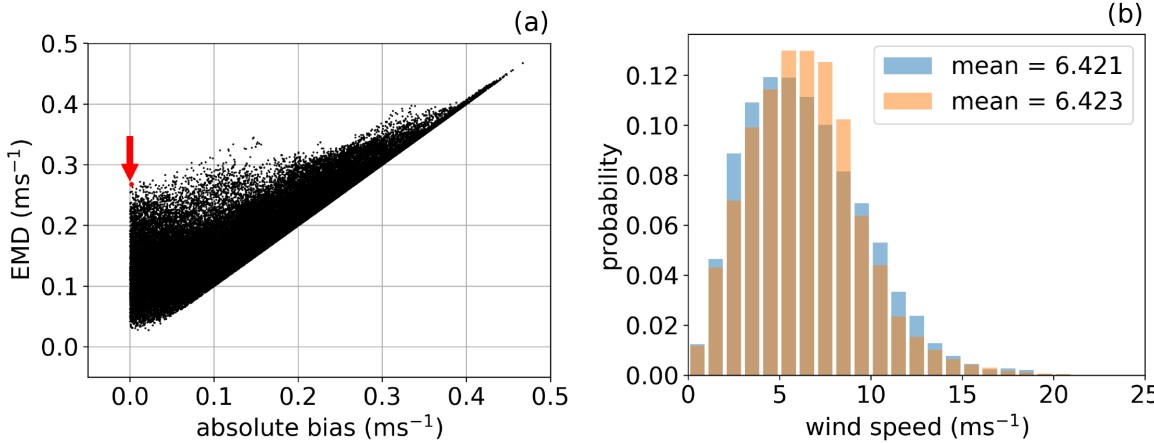

**Figure 2.** (a) Relationship between EMD values and absolute value of bias between the wind speed in two WRF model setups for all grid-points in the domain. The red dot represents an example where the EMD and the absolute value of bias are different. (b) Wind speed distributions of the two ensemble members corresponding to the red dot.

Figure 2 illustrates the differences between the EMD and the absolute value of the bias. The left panel shows the relationship between the EMD and the absolute value of bias for two WRF simulations for all points in the domain, with each dot in the plot representing one grid point. The right panel of Figure 2 shows modelled wind speed distributions for two separate grid





points to highlight a case where two differently shaped distributions can have the same mean, and using the EMD metric can identify the differences in such distributions. The values for EMD and difference in means are similar when both of the numbers are large. Therefore, using EMD instead of bias provides the greatest value when it is necessary to distinguish between two differently-shaped distributions, which might have the same means. The EMD metric has the same units as the

variable being compared. When comparing the performance of the $j$-th ensemble member to that of the baseline, we use the ratio $(\mathrm{EMD}_j - \mathrm{EMD}_B)/\mathrm{EMD}_B$; if this difference is negative, the time series of the ensemble verifies better than the baseline against the observations.

## 5    Sensitivity analysis of WRF simulations

In this section, the results from the different sensitivity experiments are presented and discussed. These are grouped into

five subsections: Section 5.1 presents the results from five different domains for a small number of experiments to see how the results differ depending on the simulated region; Section 5.2 highlights the impact of the WRF version on the wind speed results by investigating four different versions of the WRF model; Section 5.3 presents the results from a 25 simulation sensitivity study addressing the impact of the SL, PBL and LSM schemes, including changes to the surface roughness length; Section 5.4 documents the impact of other parameterizations and forcing data; and finally Section 5.5 focuses on the impact of the size of

the domain.

### 5.1    Sensitivity to geographical domain

In the initial stage of the evaluation of the WRF model setup, we designed four numerical experiments (Table 2) over five different regions in Europe (Fig. 3), mostly located near countries represented in the NEWA team. The aim was to explore the impact of using different PBL schemes either MYNN (Mellor and Yamada, 1982) or YSU (Hong et al., 2004), and the

effect of using different initialisation strategy: either using shorter or longer simulation length and exclude or include nudging. This series of numerical experiments had the main objective to clarify whether the sensitivity of the mean wind speed to these changes is similar in different geographic regions or if there were regional differences. All other settings were left fixed. The simulations were carried out with the WRF model version 3.6.1, released in August 2014. The basic WRF model setup includes the use of ERA-Interim (Dee et al., 2011) for initial and boundary conditions, NCEP optimal interpolation sea surface

temperature (SST, Reynolds et al., 2007) and 61 vertical levels. Other details are given in Table A1.

The analysis of the experiments in Table 2 showed that the largest differences arise from the choice of PBL scheme, as shown in Fig. 4 for the NW and SE domains. The left side of the figure shows the differences in annual mean wind speed at 100 m between simulations using the MYNN and YSU PBL schemes; the plots on the right side show the surface roughness length for the two domains. The results show that the regions with the largest differences coincide with the regions with particularly

large roughness length, namely forests in southern Sweden and south-western France. There, the experiment using the MYNN scheme provides wind speeds that are on average more than $0.5\,\mathrm{m\,s^{-1}}$ lower than in the experiment using the YSU scheme.




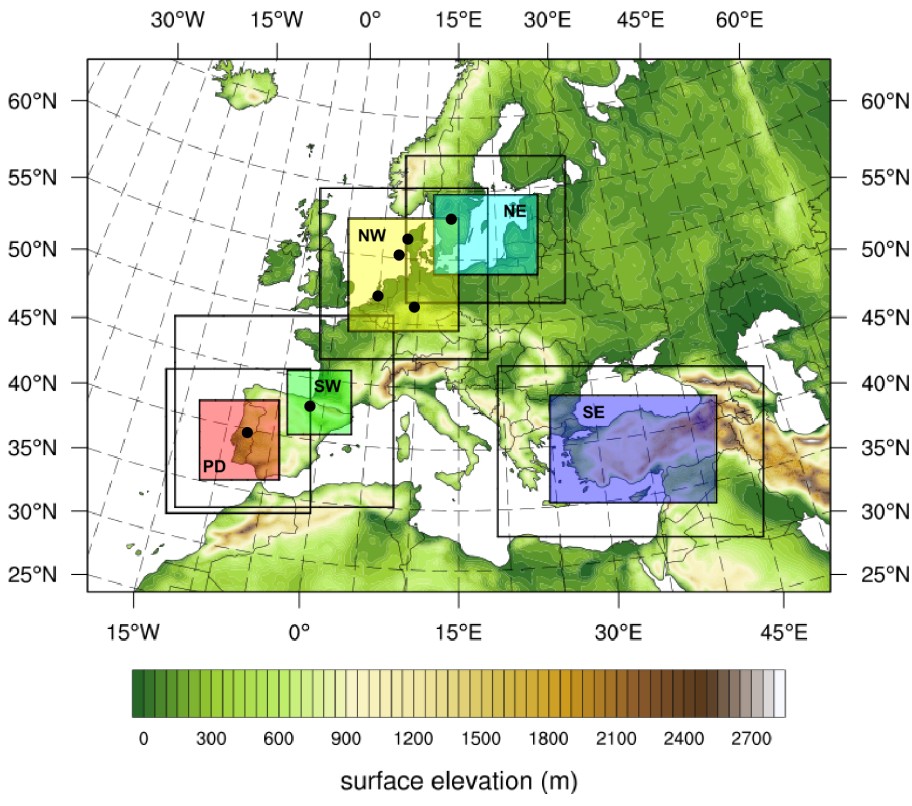

**Figure 3.** The location of the five inner domains (D3; NW, NE, SE, SW, and PD in coloured boxes) used in the geographic similarity experiments. The surface elevation of the WRF model outer domain (D1) is shown with colour; the black lines show the extent of D2. The black dots show the locations of NEWA experimental sites (Mann et al., 2017). All inner domains share the same outer domain (D1).

**Table 2.** Acronyms and relevant set up parameters of the WRF model sensitivity experiments oriented to address the influence of the geographic domain.

| experiment | PBL scheme | run | spin-up | nudging |
|---|---|---|---|---|
| | scheme | length [d] | length [h] | |
| MYNL61S1 | MYNN | 1.5 | 12 | no |
| MYNL61W1 | MYNN | 8 | 24 | yes |
| YSUL61S1 | YSU | 1.5 | 12 | no |
| YSUL61W1 | YSU | 8 | 24 | yes |

Over the sea, no significant difference is seen in the NW domain, and only slight differences (less than $0.3\,\mathrm{m\,s^{-1}}$) exist above the Atlantic and Mediterranean (SW domain).





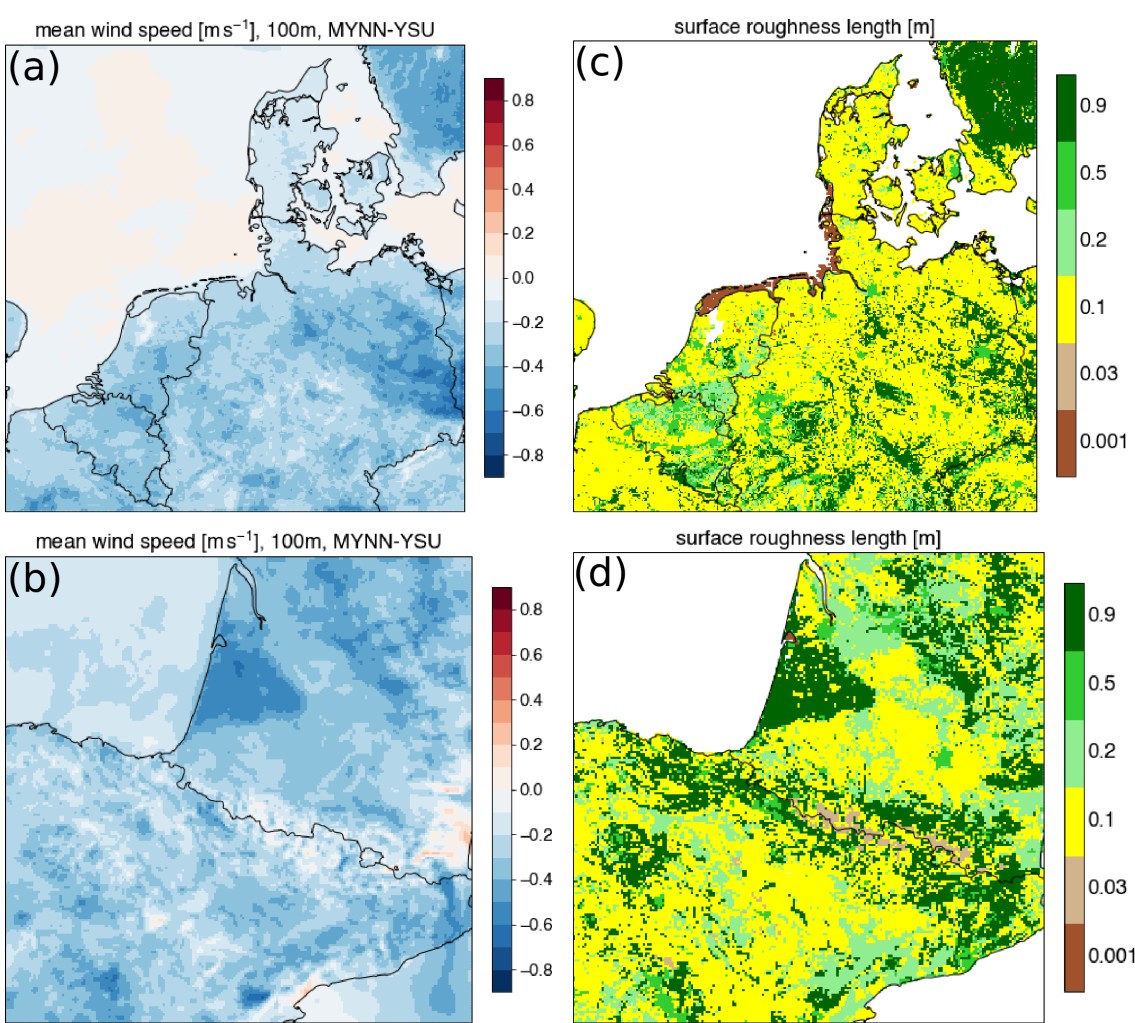

**Figure 4.** Annual mean difference in wind speed $[\mathrm{m\,s^{-1}}]$ between the MYNL61W1 and YSUL61W1 simulations for the (a) NW and (b) SW domains. Surface roughness length [m] for the (c) NW and (d) SW domains.

Similar analysis was carried out for the other three domains in Fig. 3. All five domains show the same pattern of higher wind speed at 100 m for simulations using the YSU scheme over land, with the largest differences occurring over rougher terrain (e.g. forests). Over water, the differences are much smaller, but winds simulated using the YSU scheme are slightly higher than those simulated using the MYNN scheme, but only in regions dominated by unstable stratification (e.g. the Atlantic Ocean, Mediterranean and Black Seas).

In the evaluation of the mean wind speeds from the four sensitivity experiments against the mast measurements (Fig. 5), the differences between the various experiments are small, but overall the mean winds from the MYNL61W1 run are closest to the observed value at nearly all the sites, except at FINO2. The mean statistics of the wind speed for the experiments and sites are





presented in Fig. 6. This confirms that the wind speeds from the MYNL61W1 run have: the lowest biases at Cabauw, Høvsøre, Østerild, and Risø; the highest correlation at all sites; and the lowest RMSE at all sites, except for FINO1 and IJmuiden, where the results from MYNL61W1 and YSUL61W1 are virtually the same.

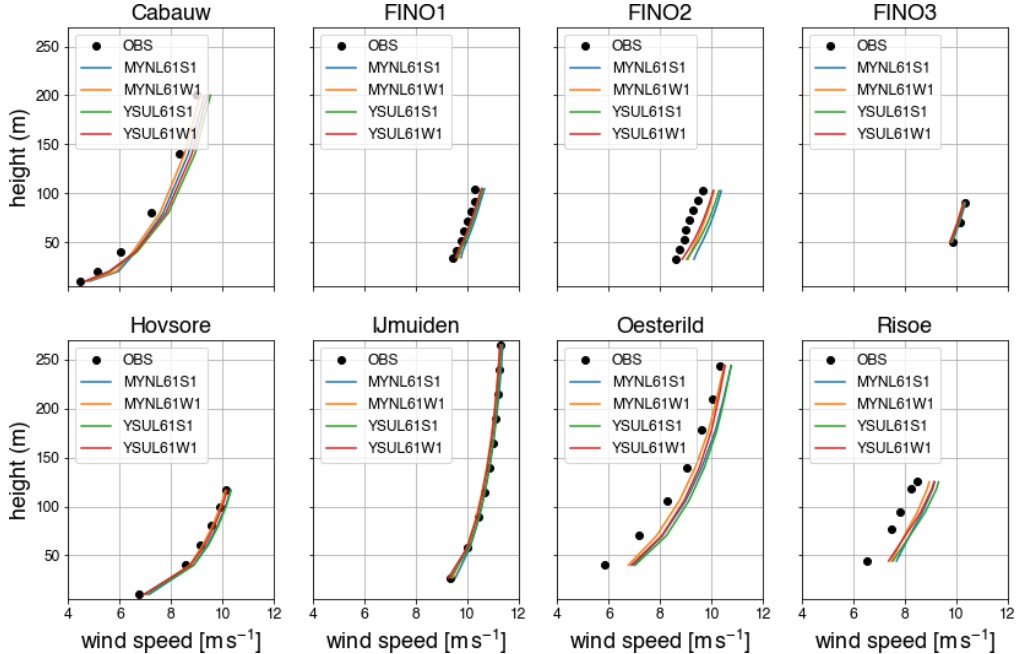

**Figure 5.** Comparison of the observed mean wind speed [m s$^{-1}$] as a function of height for the eight sites and the simulated mean wind speed from the four sensitivity runs in Table 2.

In summary, the weekly nudged simulations for both MYNN and YSU schemes result in lower biases and higher correlations at all sites. Also, because the effect on the wind speed between the two PBL schemes is nearly insensitive to the location of the domain as shown in Fig.4, this should be valid for other regions in Europe, except for regions with more complex terrain, which have not been evaluated. The weekly nudged simulation setup was chosen for the remainder of the NEWA sensitivity simulations. The use of nudging will be re-evaluated in the sensitivity experiments in Section 5.4.

### 5.2 Sensitivity to the WRF model version

The WRF model version was also tested during the sensitivity analysis to evaluate whether changing the version implied any difference with respect to the baseline configuration described above. At the time of the development of this work, the latest version was WRF version 3.9.1, released in August 2017. The simulations using versions WRFV3.8.1 and WRFV3.9.1 for the same NW domain and the same model setup as in MYNL61W1 were carried out. The results for WRFV3.6.1 and WRFV3.8.1 are presented in Fig. 7; the results for WRFV3.9.1 (not shown) are almost identical to those from WRFV3.8.1.



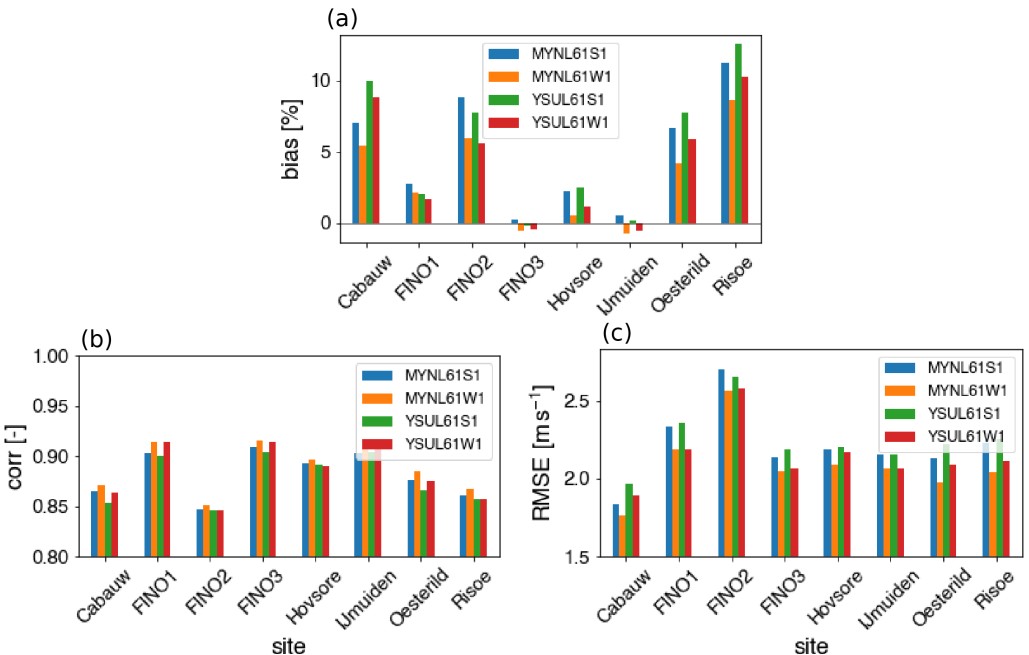

**Figure 6.** Evaluation statistics: (a) bias [%], (b) correlation [-], and (c) RMSE [m s$^{-1}$] for the simulated wind speed for the eight sites and four sensitivity experiments in Table 2.

The wind speed from simulations using WRFV3.8.1 presented an increased bias compared to observations for all sites and most levels, except for FINO1 and FINO2, which suffer from flow distortion and wind farm effects. The increased bias was traced back to changes in two important equations in the MYNN SL and PBL scheme. The first is the scalar roughness length over water, which was changed from the formulation in Fairall et al. (2003) to that in Edson et al. (2013), thus affecting the wind
5 speed over the ocean. The second is a change in the definition of the mixing length (Olson et al., 2016). Both of these options could be customised and set as in the baseline configuration. Results from such a setup are labelled "WRFV3.8.1_MOD" in Fig. 7. Some of the previous characteristics of the profile are restored after these changes, and the simulation using WRFV3.8.1 with modifications improved the RMSE for all sites at ∼ 100 m (not shown). These changes are consistent with those found by Yang et al. (2017) for the MYNN scheme. Although above ∼ 100 m, the simulation using the modified WRFV3.8.1 gives
10 lower mean wind speeds than WRFV3.6.1 at all sites, we consider nonetheless that this differences is less relevant and based on the improvements of the scheme "WRFV3.8.1_MOD", this set up was selected as the baseline, named "BASE" hereon. The WRF model setup of this baseline is summarised in Table A1. Unless explicitly labelled otherwise, when referring to the MYNN option in the remainder of this work we mean the modified version of the scheme. The unmodified MYNN PBL will be referred to as MYNN*.



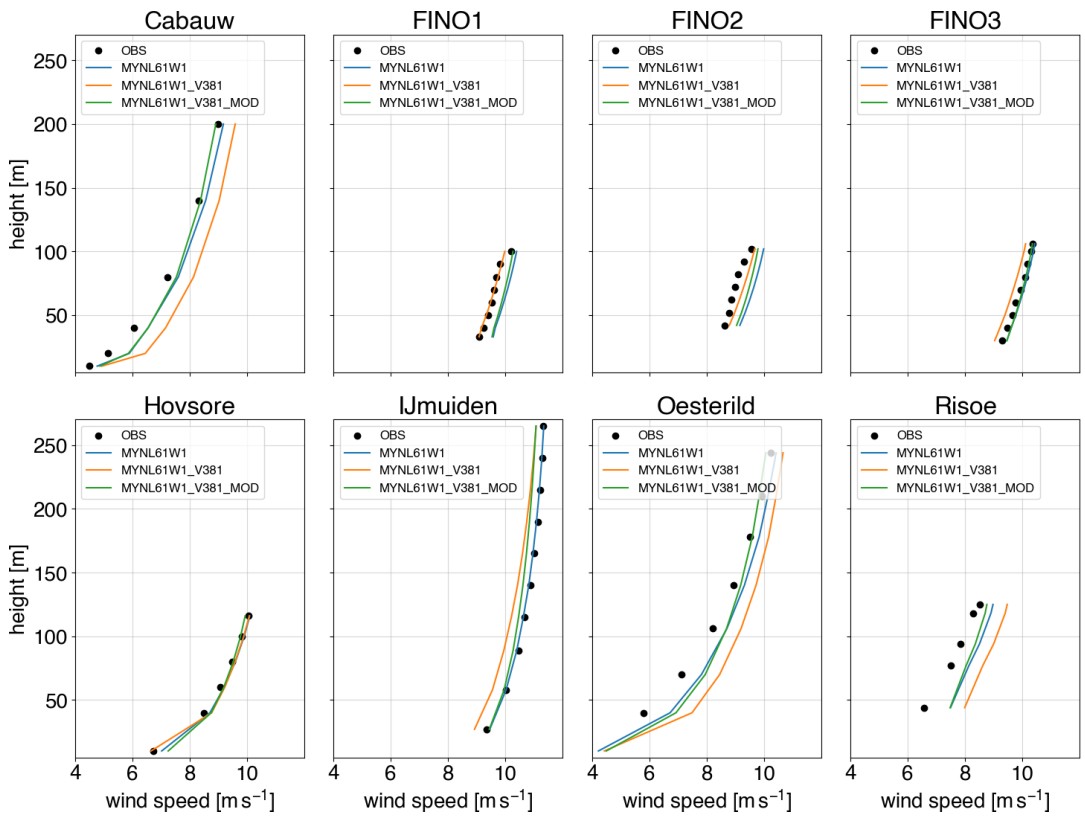

**Figure 7.** Comparison of the mean observed wind speed [m s$^{-1}$] as a function of height for the eight sites and the mean wind speed in the simulations using WRFV3.6.1, WRFV3.8.1 and the modified version of WRFV3.8.1_MOD.

## 5.3 Effect of surface and planetary boundary layer and land surface model

The first series of sensitivity studies tested the sensitivity of the near surface wind to various combinations of LSM, PBL, SL schemes and the specification of surface roughness length. The schemes tested are listed in Table 3.

The large number of schemes and their potential combinations led to a large number of possible combinations. In this work a total of 25 different combinations were tested, listed in Table 3, including changes in parameters in the schemes themselves. We also included the simulation using the unmodified MYNN PBL (labelled MYNN*, see Section 5.2) scheme. Our original table of sensitivity experiments contained many other LSM/PBL/SL combinations, but some of these suffered from diverse technical issues and did not complete the runs. In some cases, small adjustments were needed, that is the fractional sea-ice option had to be turned off when using the MM5 or MO surface layer schemes.

An important aspect of the LSM/PBL/SL sensitivity studies was the use of an alternative look-up table for the surface roughness length as a function of land-use class. A custom NEWA lookup table was created since many values used in the WRF-distributed tables do not match the aerodynamic characteristics of European vegetation, especially over forests (Hahmann





**Table 3.** Overview of the ensemble of WRF model simulations varying LSM/PBL/SL scheme and vegetation lookup table carried out for the NW domain. The names of the schemes are: LSM: NOAH (Tewari et al., 2004), RUC (Benjamin et al., 2004), PX (Noilhan and Planton, 1989), SLAB (Dudhia, 1996), NOAH-MP (Niu et al., 2011); PBL/SL: YSU (Hong et al., 2006), MYJ (Mellor and Yamada, 1982), MYNN (Nakanishi and Niino, 2006), modified MYNN see Section 5.2), ACM2 (Pleim, 2007), MM5 (Jiménez et al., 2012), M-O surface layer (Janjic and Zavisa, 1994). The number in parenthesis is the respective namelist value in the WRF model configuration file. The "simulation code" is explained in Appendix A2.

| run name | simulation code | LSM (#) | PBL (#) | SL (#) | veg table |
|---|---|---|---|---|---|
| BASE | EES81_2551040004 | NOAH (2) | MYNN (5) | MYNN (5) | NEWA |
| MYNN* | EES81_2550040004 | NOAH (2) | MYNN*[a] (5) | MYNN*[a] (5) | NEWA |
| MM5 | EES81_2511040004 | NOAH (2) | MYNN (5) | MM5 (1) | NEWA |
| MO | EES81_2521040004 | NOAH (2) | MYNN (5) | M-O (2) | NEWA |
| MYJ-MO | EES81_2220040004 | NOAH (2) | MYJ (2) | M-O (2) | NEWA |
| YSU-MM5 | EES81_2110040004 | NOAH (2) | YSU (1) | MM5 (1) | NEWA |
| RUC | EES81_3551040004 | RUC (3) | MYNN (5) | MYNN (5) | NEWA |
| RUC-WRF | EES81_3551040004_A | RUC (3) | MYNN (5) | MYNN (5) | WRF |
| RUC-MO | EES81_3521040004 | RUC (3) | MYNN (5) | M-O (2) | NEWA |
| RUC-YSU-MM5 | EES81_3110040004 | RUC (3) | YSU (1) | MM5 (1) | NEWA |
| RUC-ACM2-PX | EES81_3770040004 | RUC (3) | ACM2 (7) | ACM2 (7) | NEWA |
| PX-ACM2-PX | EES81_7770040004 | PX (7) | ACM2 (7) | ACM2 (7) | NEWA |
| PX-ACM2-MM5 | EES81_7710040004 | PX (7) | ACM2 (7) | MM5 (1) | NEWA |
| SLAB | EES81_1551040004 | SLAB (1) | MYNN (5) | MYNN (5) | NEWA |
| SLAB-MYJ-MO | EES81_1220040004 | SLAB (1) | MYJ (2) | M-O (2) | NEWA |
| SLAB-YSU-MM5 | EES81_1110040004 | SLAB (1) | YSU (1) | MM5 (1) | NEWA |
| SLAB-ACM2-PX | EES81_1770040004 | SLAB (1) | ACM2 (7) | ACM2 (7) | NEWA |
| NOAHMP | EES81_4550040004 | NOAH-MP (4) | MYNN (5) | MYNN (5) | NEWA |
| NOAHMP2[b] | EES81_4550040004_B | NOAH-MP (4) | MYNN (5) | MYNN (5) | NEWA |
| NOAHMP-WRF | EES81_4550040004_A | NOAH-MP (4) | MYNN (5) | MYNN (5) | WRF |
| NOAHMP-MYJ-MO | EES81_4220040004 | NOAH-MP (4) | MYJ (2) | M-O (2) | NEWA |
| NOAHMP-YSU-MM5 | EES81_4110040004 | NOAH-MP (4) | YSU (1) | MM5 (1) | NEWA |
| ANNZ0[c] | EES82_2551040004 | NOAH (2) | MYNN (5) | MYNN (5) | WRF[d] |
| ANNZ0N[c] | EES82_2551040004_A | NOAH (2) | MYNN (5) | MYNN (5) | WRF |
| AGGZ0[c] | EES83_2551040004 | NOAH (2) | MYNN (5) | MYNN (5) | NEWA |

[a] is the unmodified MYNN scheme, [b] uses opt_sfc = 2 in the NOAH-MP scheme. [c] differs in surface roughness, see text for details. [d]WRF table, but low roughness for tidal zone.

et al., 2015; Floors et al., 2018a). The new lookup table was created by polling wind energy resource assessment experts from the NEWA consortium. Both the new and old values of surface roughness length for each roughness class are shown in Table 4.





**Table 4.** Vegetation look-up table for the surface roughness length as a function of the USGS land use category (Anderson et al., 1976) in the NEWA and default NCAR WRF model configuration. Only values changed from default are shown.

| USGS type | land-use land cover class | $z_0$ NEWA [m] | $z_0$ WRF orig range [m] |
|---|---|---|---|
| 2 | Dryland Cropland and Pasture | 0.10 | 0.05–0.15 |
| 3 | Irrigated Cropland and Pasture | 0.10 | 0.02–0.10 |
| 4 | Mixed Dryland/Irrigated Cropland and Pasture | 0.10 | 0.05–0.15 |
| 5 | Cropland/Grassland Mosaic | 0.10 | 0.05–0.14 |
| 7 | Grassland | 0.10 | 0.10–0.12 |
| 8 | Shrubland | 0.12 | 0.01–0.05 |
| 9 | Mixed Shrubland/Grassland | 0.12 | 0.01–0.06 |
| 11 | Deciduous Broadleaf Forest | 0.90 | 0.5 |
| 12 | Deciduous Needleleaf Forest | 0.90 | 0.5 |
| 13 | Evergreen Broadleaf Forest | 0.90 | 0.5 |
| 14 | Evergreen Needleleaf Forest | 0.90 | 0.5 |
| 15 | Mixed Forest | 0.50 | 0.20–0.50 |
| 17 | Tidal zone[a] | 0.001 | 0.20 |

[a] Originally called "Herbaceous Wetland" in the default WRF model vegetation table.

Some of the larger changes include, "Herbaceous Wetland", which has an original value of $z_0 = 0.20$ m, but in the NEWA regional represents the tidal zone in coastal Holland, Germany, and Denmark, which is much smoother (Wohlfart et al., 2018), and thus was changed to $z_0 = 0.001$ m. The forest classes were also significantly changed, with the $z_0 = 0.50$ m value in the default table being changed to $z_0 = 0.90$ m, which is more representative of forests in, for example Sweden (Dellwik et al., 2014). The new roughness values should be considered only as estimates and as such there might be some limitations in the representation of the roughness length, they are nevertheless much more realistic than the default ones. The experiments using the standard vegetation tables are labelled WRF vegetation in Table 3.

All NEWA setups use a constant value of surface roughness and have no annual cycle, except for two of the setups (ANNZ0 and ANNZ0N) that have annual cycle according to the default WRF vegetation table (except for the tidal zone). In WRF, the seasonality of the surface roughness length is controlled by the value of the green vegetation fraction (Refslund et al., 2014) and applies mostly for cropland classes in Table 4. The annual cycle of green vegetation fraction does not change from year to year and is spatially inconsistent with the ESA-CCI land-use dataset used in the NEWA simulations. Therefore, because of inherent uncertainties, in NEWA we have chosen to use a single constant value of roughness for land-use and land cover class. Also, since the WRF wind climatologies will be further downscaled, as described in Dörenkämper et al. (2020), using a constant value of surface roughness facilitates the process. Another roughness-related experiment that was included, AGGZ0, uses the sub-tiling option for NOAH (Li et al., 2013), with the NEWA vegetation table. The sub-tiling option generates more realistic




values of surface roughness length in areas of mixed vegetation, which could reduce the biases in wind speed (Santos-Alamillos et al., 2015).

The vertical profiles of mean wind speed for all the LSM/PBL/SL sensitivity experiments for four of the eight evaluation sites are shown in Fig. 8 as an example. It is difficult to distinguish between the results of the various setups, but generally, the setups using the MYNN PBL scheme (in blue) tend to have lower wind speeds, which are often closer to the observed values over the sea. The spread in the wind speed among the simulations excluding outliers generally increases with height reaching around $1\,\mathrm{m\,s^{-1}}$ at $100\,\mathrm{m}$ over the sea and around $1.5\,\mathrm{m\,s^{-1}}$ over land.

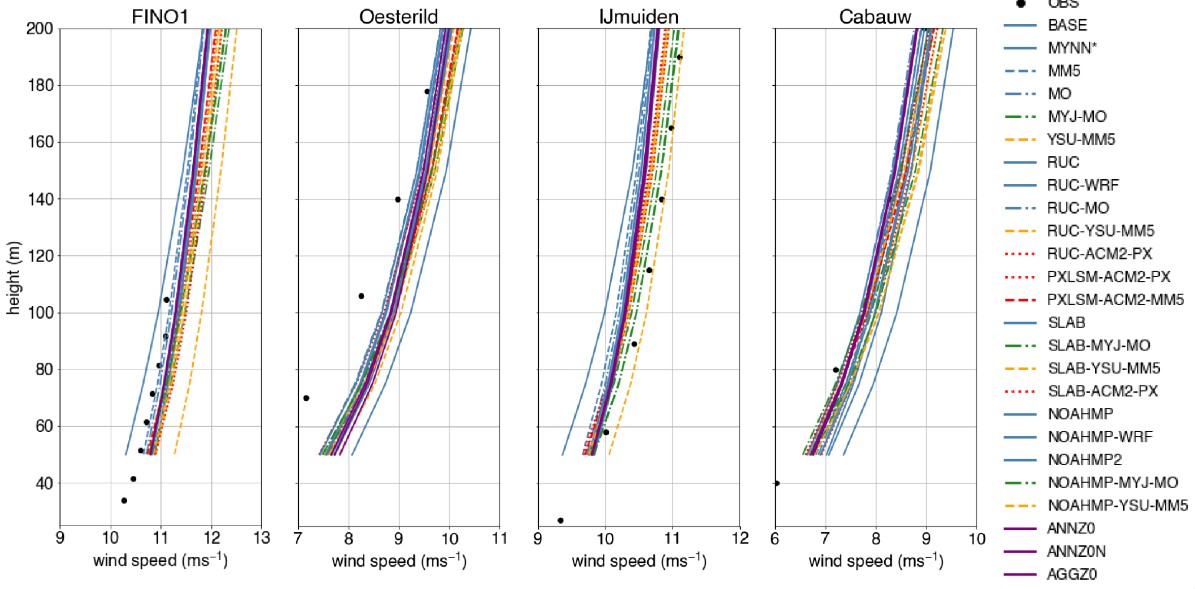

**Figure 8.** Mean wind speed $[\mathrm{m\,s^{-1}}]$ simulated by the various experiments in the LSM/SL/PBL ensemble as a function of height for four sites: FINO1, Øesterild, IJmuiden and Cabauw. The observed values are shown as black dots; simulations using the same PBL scheme are given the same colour, except for the simulations with modified surface roughness.

To facilitate the intercomparison among the ensemble members, we computed the evaluation metrics of the wind speed for each simulation (Fig. 9 and 10). The left panel of Fig. 9 shows the relative model bias at all the sites. It shows that the bias shows a certain relation with the site, expressed in this figure as consistent colours for each column. Additionally, the characteristic of the bias relates most directly to the type of site, i.e. slightly negative bias over the sea and positive bias over land. The latter is likely a consequence of deficient representation of the land characteristics around each site, since they are independent of the LSM/PBL/SL used. Some other general patterns are that the simulations using the MYJ scheme tend to have largest absolute biases, except at FINO3, and that the YSU-MM5-RUC simulation is an outlier, whose results differ from other setups, typically being among the worst of setups for any station, which is also evident in the vertical profiles at FINO1 and IJmuiden in Fig. 8.

To better quantify the differences between the simulations, the right panel of Fig. 9 shows the absolute difference in the relative bias from the "BASE" simulation as defined in Section 4. Negative numbers show a decrease in bias, which will point




to a more accurate simulation. From these values some conclusions can be drawn. First, the differences of the simulations are quite modest, with a maximum of 8.9 % in the MYNN* simulation at Cabauw and −1.9 % in several simulations at various sites. Second, no simulation is capable of improving or degrading the bias statistics at all of the sites: the MYJ-MO-SLAB simulation improves the bias at three of the eight sites, while the MO simulation performs better at four of the eight sites. The

unmodified MYNN scheme considerably degrades the simulations at six of the eight sites. The latter supports our decision to use the modified version of the MYNN scheme as baseline. It is relevant to note that the changes in $z_0$ cause minor variations in the biases (see members ANNZ0, ANNZ0n, AGGZ0), however the sites are located in regions with vegetation classes that did not change significantly from the WRF model standard table.

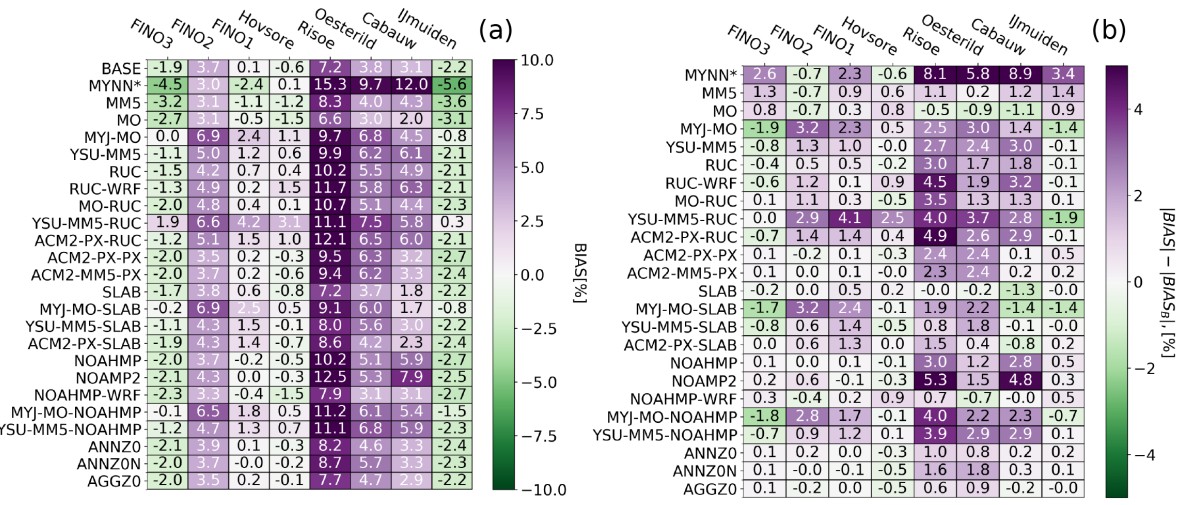

**Figure 9.** (a) Biases [%] and (b) changes in the biases from the BASE simulation [|BIAS| − |BIAS_B|, %] between the observed and simulated wind speed at the eight sites and the various sensitivity studies in the LSM/SL/PBL ensemble (Table 3).

Fig. 10 provides further information about the sensitivity tests based on the EMD metric defined in Section 4 to evaluate the
shape of the wind speed distributions. As with the bias, the EMD metric shows that the largest differences in total error are related to the site location, with the best model performance at Høvsøre, with EMD between (0.6–3.2 %), while at Risø the results fall between (6.8–15.3 %). Particularly interesting is the comparison between the two metrics, bias and EMD, since for most setups and stations the values of EMD and bias are similar, especially when the model results are significantly different from the observations. However, for FINO1, if only the bias was analysed, it could be argued that the base setup represents
the observed distribution perfectly. However, the EMD shows that it is not the case and overall performance of the model is comparable to IJMuiden. The EMD allows us to identify cases where the change in model setup improves only the mean value of distribution, as opposed to the similarities of the whole distribution. For instance, if only the biases were analysed, it could be argued that MYJ-MO is significantly better than the base in FINO3 and IJmuiden, while the EMD results show that there is only a modest improvement over the base at IJmuiden and a worse result at FINO3. Similar conclusions can be carried





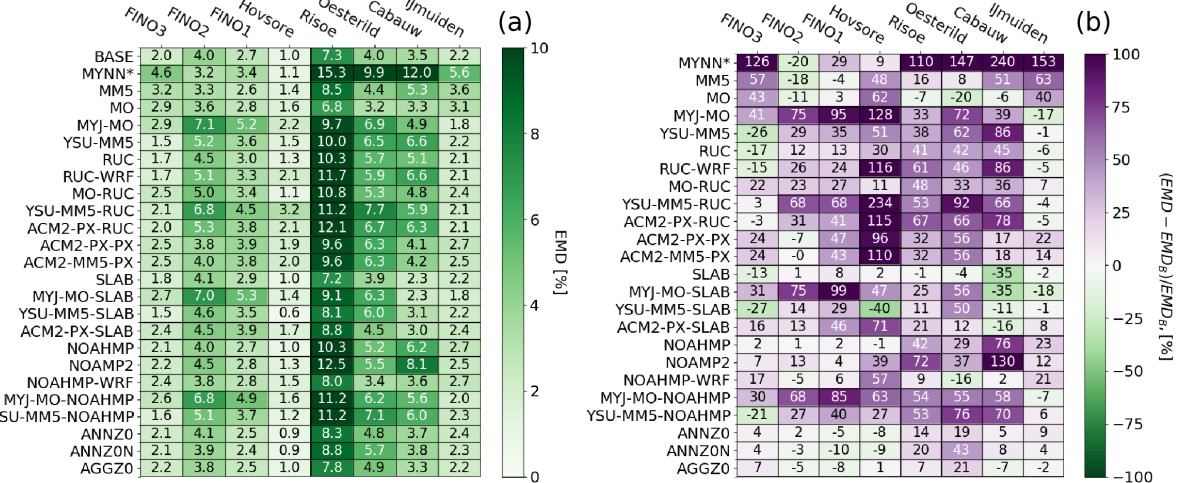

**Figure 10.** (a) EMD relative to the observed wind speed [%] and (b) relative change in the EMD from the BASE simulation [(EMD − EMD_B)/EMD_B, %] between the observed and simulated wind speed at the eight sites and the various sensitivity studies in the LSM/SL/PBL ensemble (Table 3).

out about other runs using the MYJ-MO PBL scheme and YSU-MM5-RUC run. However, as with the bias, no simulation improves the EMD for all sites, and very few simulations improve the EMD metric at all, especially for the land sites. The SLAB simulation significantly improves EMD at two sites, (>5 %, relative to the base), while the MO simulation significantly improves the EMD at four of the eight sites.

Based on these results, the MO simulation, which uses the NOAH LSM, MYNN PBL scheme and MO surface layer scheme (MO) was selected as the configuration for the NEWA production run.

### 5.4 Other sensitivity experiments

A second set of sensitivity experiments was carried out to identify other factors that could potentially be important for the simulation of wind speed within the WRF model. These experiments are listed in Table 5 and can be grouped into three main

categories. First, we tested the impact of various initial and boundary conditions, by using ERA-Interim, MERRA2 and FNL fields as forcing. The effect of various sources of sea surface temperature (SST) was also tested. In a second set, we tested other model dynamics including the effect of spectral versus grid nudging, enlarging the lateral boundary zone, changing the wavelength of the minimum spectral nudging length and enabling 2-way nesting. The third set of experiments tested other model physics not related to the surface and PBL, that is radiation, cumulus convection and explicit moisture schemes.

Figure 11 shows the bias and bias differences compared to the BASE simulation for the additional set of sensitivity experiments. In contrast to the LSM/PBL/SL ensemble members, these simulations provide results that are very similar to the BASE simulation, except for the CAMRAD simulation, which was run replacing the usual RRTM radiation scheme (Mlawer et al.,



**Table 5.** Overview of other sensitivity experiments carried out. The meaning of the simulation code is explained in Table A2 in the appendix.

| run name | simulation code | changes to BASE run |
|---|---|---|
| *Initial, boundary conditions and SST* | | |
| ERAI | IIS81_2551040004 | ERA-Interim (Dee et al., 2011) forcing |
| MERRA2 | MMS81_2551040004 | MERRA2 (Gelaro et al., 2017) forcing |
| FNL | FFS81_2551040004 | FNL (NCAR, 2000) forcing |
| ERA5SST | EEE81_2551040004 | ERA5 SST |
| HRSST | EEH81_2551040004 | NOAA HRSST (Gemmill et al., 2007) |
| OISST | EEO81_2551040004 | NOAA RTGSST Reynolds et al. (2010) |
| *Model dynamics* | | |
| NUDPAR | EES81_2551040004_A | lower wavelength in nudging |
| LRELAX | EES81_2551040004_B | larger relaxation zone |
| TWOWAY | EES81_2551040004_C | two-way nesting |
| NUDD3 | EES81_2551040004_J | spectral nudging D1, D2, D3 |
| GNUD1 | EES81_2551040004_I | grid nudging D1 |
| GNUD3 | EES81_2551040004_H | grid nudging D1, D2, D3 |
| *Other physics* | | |
| RAD3S | EES81_2551040004_D | radiation $\Delta t = 3\,\mathrm{s}$ in all domains |
| RAD12S | EES81_2551040004_E | radiation $\Delta t = 12\,\mathrm{s}$ in all domains |
| FASTRA | EES81_2551040024 | fast RRTMG code |
| CAMRAD | EES81_2551040003 | CAM radiation (Collins et al., 2004) |
| CUG-F | EES81_2551040304 | Grell-Freitas (Grell and Freitas, 2014) CU scheme |
| THOMP | EES81_2551080004 | Thompson cloud physics (Thompson et al., 2012) + icing |

1997) with the CAM parameterisation scheme (Collins et al., 2004). Switching the source of initial and boundary conditions to ERA-Interim, MERRA2 or FNL has very small impact to the bias ratios. The only significant change is at Cabauw, where the biases are increased by 1–2 % by using any of these three other forcing data. Changing the source of SST has an insignificant effect for all of the offshore masts. Most of the changes to the dynamic settings have very small consequences to the bias.

5    The only significant change is the use of grid nudging in all three WRF domains, simulation GNUD3. The biases increase in six of the eight sites by 0.2–2.0 % when this setting is activated, probably because of the slow down of the winds in the ERA5 reanalysis over land (see Fig. 9 of Dörenkämper et al., 2020). Interestingly, using this setting significantly increases the correlation between the simulated and observed time series (not shown), but at the expense of increased biases.

Similarly, the relative EMD for this set of sensitivity experiments was calculated and the results are shown in Figure 12.

10   The left panel shows the EMD relative to the observed wind speed and right panel the relative improvement of this metric with respect to the baseline. The conclusions about the usefulness of EMD metric for FINO1 and Høvsøre apply here as well.





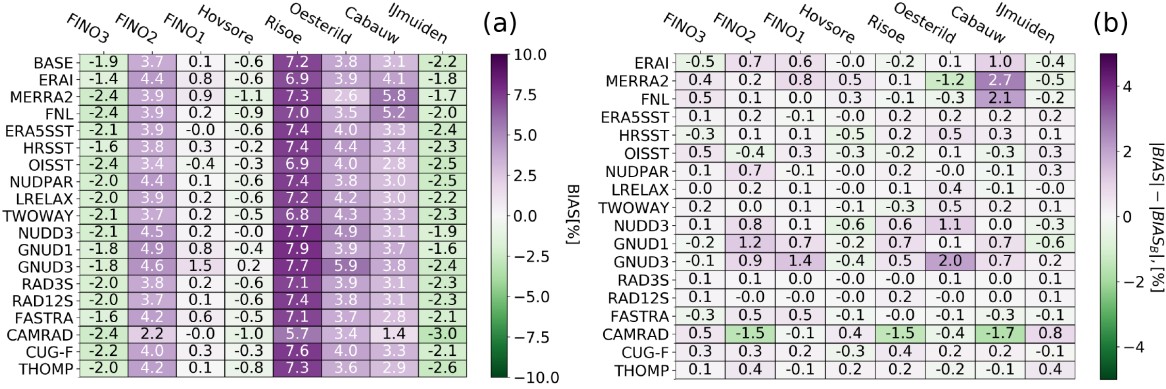

**Figure 11.** (a) Biases [%] and (b) relative change in the biases from BASE simulation [|BIAS| − |BIAS_B|, %] between the observed and simulated wind speed at the eight sites and the various sensitivity studies in the non-PBL ensemble (Table 5).

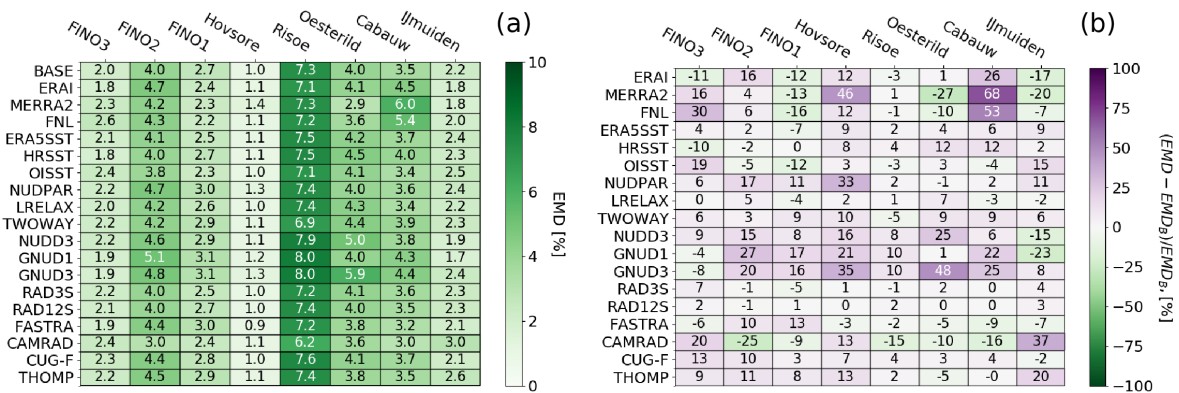

**Figure 12.** (a) EMD relative to the observed wind speed [%] and (b) relative change in the EMD from the BASE simulation [(EMD − EMD_B)/EMD_B, %] between the observed and simulated wind speed at the eight sites and the various sensitivity studies in the non-PBL ensemble (Table 5).

For instance, at FINO1 the ERAI and MERRA2 runs show an increase in bias, while the EMD values show that these runs actually have more similar wind speed distribution to observations than the BASE. Otherwise, the EMD metric confirms the conclusions described earlier about the small impact of all of these changes and the relative decrease in quality when using grid-nudging (GNUD3).

5    In conclusion, many other changes to the WRF model settings have inconsequential effects to the simulation of the wind speed at wind turbine hub height. The change in radiation parameterisation has a small effect in relation to the BASE simulation. Unfortunately, we did not run a simulation with the MO and CAMRAD together, so it is not possible to assess if that simulation would have been more accurate. Because the effect is small and the CAM radiation parameterisation (Collins et al., 2004) is





more expensive in terms of computational resources, it was ultimately decided to keep the NOAH-MYNN-MO setup as the choice for the NEWA production run.

## 5.5 Domain size

An additional decision to be made regarding the NEWA mesoscale simulations was the domain configuration; that is using a single large domain or several small domains to cover Europe. From a pure computational perspective, one single domain is more efficient, because the WRF model code scales better with larger domains (Kruse et al., 2013) and there is only data from one domain to post-process. However, the output files are very large and the simulation needs to be completed before post-processing can begin. The limiting factor here is the scratch space available at modern HPC systems that is typically not more than 100 TB. Furthermore, large areas outside of the region of interest (the NEWA domain, see Section 1) would be simulated, that is parts of the Atlantic Ocean, the Norwegian Sea and non-EU countries in Eastern Europe, thus a substantial amount of computational resources would be wasted. Apart from these technical questions, it was unknown how the domain size influences the quality of the simulated fields.

To study the sensitivity of the simulated wind speed, we carried out simulations for three differently sized domains over the North Sea using the same setup and resolution as in Section 5.1. The number of grid points in the inner domain in these simulations are: small (SM) 121 × 121, medium (MD) 241 × 241 and large (LG) 481 × 481, which correspond to square domains with edge lengths on the WRF model projection equivalent to 360 km, 720 km, and 1440 km, respectively. The three domains are centred at the same coordinates and only differ by the number of grid points. The size of the boundary zone, in grid cells, between D1 and D2 and D2 and D3 is kept the same. Two sets of WRF model simulations were done for each of the three domains with daily and weekly runs, analogous to MYNL61S1 and MYNL61W1 in Table 2. For evaluating the results of the simulations we use the same data as in Section 5.1, but only six of the masts are contained within the SM domain.

Figure 13 shows the biases between the various WRF model simulations and the observations for the 6 sites. The biases from all simulations are summarised as follows: for five out of six sites the MYNL61S1/LG simulation have the largest biases, and for all sites the MYNL61W1/SM simulation has the smallest biases. Similar results (not shown) emerge for the correlation and the RMSE. Particularly at Høvsøre, the bias decreases from 2.9 to 1.6 to 0.6 in the MYNL61W1 LG, MD, and SM simulations, respectively.

In conclusion, for this region, biases in mean wind speed are influenced by the size (and possibly also location, not shown) of the domain, smaller domains have generally lower wind speeds and thus lower biases. This effect is most pronounced in the week-long and "nudged" simulations. Time correlations decrease (and RMSE increase) with increasing domain size and integration time.

The results from these experiments guided the design of the NEWA domains for the production run. Instead of a single, or a few, very large domains, we chose to conduct the simulations in a rather large number of medium-sized domains. While generating different time series, overlapping areas in simulations generally show similar wind climates (Witha et al., 2019, Section 2.1.3). We decided, however, against very small domains. In terms of accuracy they would probably perform better than our chosen configuration, but most countries would be covered by multiple domains, which would face overlapping issues.





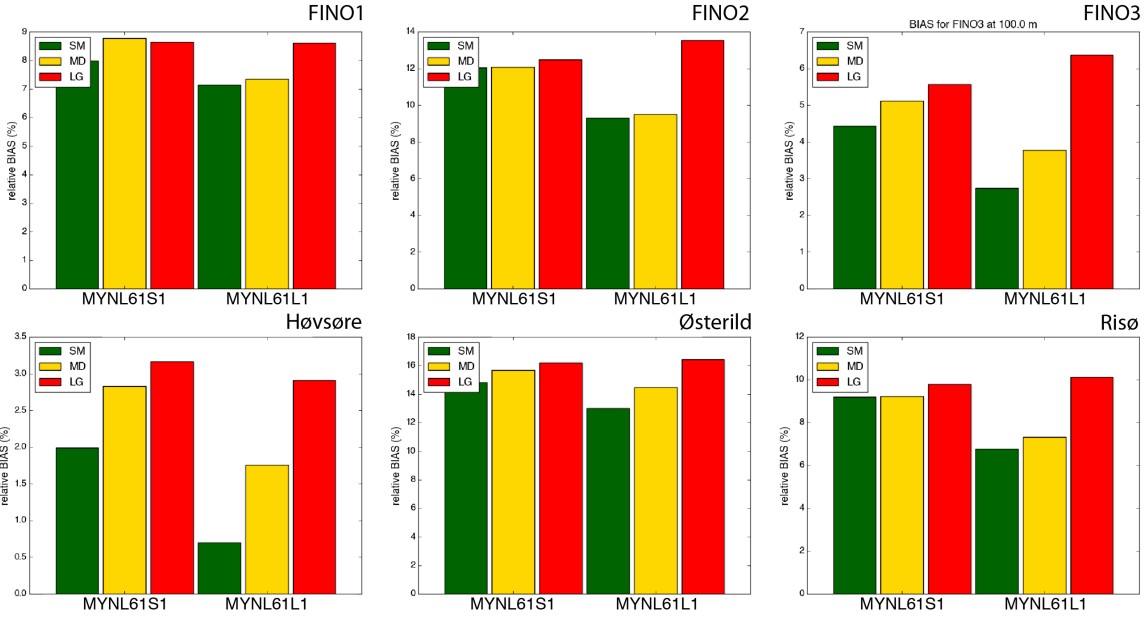

**Figure 13.** Relative bias [%] in annual mean wind speed for each domain size and simulation length for FINO1, FINO2, FINO3 (top) and Høvsøre, Østerild and Risø (bottom).

It was desired that each country should be covered by only one domain to avoid these issues. The final domain configuration, which is presented in this paper's companion (Dörenkämper et al., 2020), fulfils this requirement for all countries except Norway, Sweden and Finland which are so elongated that a correspondingly large domain would be detrimental to the accuracy of the results.

## 5.6 Summary of the sensitivity experiments

A long list of sensitivity studies were carried out to identify the ideal configuration for the NEWA production run. Here is a summary of the findings:

1. In the initial sensitivity experiments, the largest differences in annual mean wind speed at 100 m a.g.l. vs height in figures and AGL/AMSL in Table 2 are between simulations using two PBL schemes (MYNN and YSU) and coincide with regions of high surface roughness in all domains over Europe. Over the sea, the differences could be traced to differences in atmospheric stability, but were modest.

2. The weekly simulations using spectral nudging tend to perform better (lower biases and higher correlations) for the eight sites in northern Europe. The simulation using the MYNN scheme in the WRF model version 3.6.1 outperformed the simulations using the YSU scheme in this region.





3. The use of the WRF model V3.8.1 and the MYNN scheme increased the biases compared to observations at nearly all sites and most levels. A couple of settings, mynn_mixlength=0 and COARE_OPT=3.0, turn the MYNN scheme nearly back to the conditions in the WRF model version 3.6.1. However, above 100 m a.g.l. the modified MYNN scheme gives lower wind speeds than the one in WRF V3.6.1 at all sites.

4. A series of 25 experiments varying the land, PBL and surface layer scheme (LSM/PBL/SL) shows a spread in the mean wind speed of about $1\,\mathrm{m\,s^{-1}}$ over the ocean and $1.5\,\mathrm{m\,s^{-1}}$ over land. When comparing to wind speed observations, most LSM/PBL/SL ensembles show negative biases over the ocean (except for FINO2) and positive over land, which are more consistent between sites than LSM/PBL/SL combinations. This likely reflects misrepresentation of the land surface around each site than deficiencies in the LSM/PBL/SL schemes themselves.

5. Changes to the WRF model lookup table for surface roughness length have large consequences for the simulated wind speed, but is nearly invisible to the evaluation against the tall masts because these are located in areas away from those impacted by the changes.

6. The use of the EMD metric helps clarify the comparison of the improvements between the various LSM/PBL/SL and the baseline simulation, especially if the bias is small. No simulation improves the EMD for all sites, and very few simulations improve the EMD metric at all, especially for the land sites. However, the MO simulation, which uses the NOAH, MYNN and MO surface layer schemes, improves the results $> 1\,\%$ at four of the eight sites and was finally chosen as the physical model configuration for the NEWA production run.

7. A set of additional sensitivity experiments, change source of forcing data, SST, dynamic options and other physical parameterisations, shows smaller changes from the baseline simulation than the various LSM/PBL/SL experiments. Nearly all the changes have inconsequential effects to the simulation of the wind speed at wind turbine hub height. Only the simulation using the CAM radiation scheme showed improvements over the RRTMG scheme used in the baseline experiment. However, it was concluded that the modest improvements were not worth the additional expense of running this scheme in the production run.

8. A final set of experiments testing the effect of the size of the domain on the simulated wind speed error statistics showed that for a domain centred over Denmark, the simulations using the smaller domain have lower wind speeds which compare better to measurements and time correlations decrease with domain size. It is however unclear if this is a consequence of the domain size itself or the location of the main inflow boundary to the domain in the simulations.

## 6 Discussion and outlook

In the companion paper (Dörenkämper et al., 2020), we document the final model configuration and how we computed the final wind atlas, including a detailed description of the technical and practical aspects that went in to running the WRF simulations and the downscaling using the linearised microscale model WAsP (Troen and Petersen, 1989). This second paper also shows a



comprehensive evaluation of each component of the NEWA model-chain using observations from a large set ($n = 291$) of tall masts located all over Europe. We conclude that the NEWA wind climates estimated by WRF and WAsP are significantly more accurate than using ERA5 reanalysis data.

As with any modelling study, some questions remain unresolved simply because of the expensive nature of the numerical
experiments. For convenience and simplicity, we separated the sensitivity experiments dealing with LSM/PBL/SL and the other parameterisations changing only one scheme or parameter at a time. However, the experiment using the CAM radiation scheme had better verification statistics than the other simulations, but it was not tested using the final LSM/PBL/SL combination. Therefore, a better way to go in this process would be to sequentially go through the changes and evaluation in a sequential way. But the number of ensemble members can rapidly become unmanageable. Algorithms in this direction are currently being
applied for tuning Earth System Models (Li et al., 2019) and could perhaps be evaluated to best optimal WRF setups for different applications, not just wind resource assessment.

The dependence of the WRF model simulation of the wind climate on the size and location of the computational domain also remains unresolved. Smaller domains in the WRF simulation tend to have smaller wind speed biases and higher correlations compared to tall mast observations, but it was unclear if this was really a result of the size of the domain or rather the location of
the boundaries in relation to the large-scale flow. More numerical experiments should be carried out to identify these potential interactions.

Finally, it would have been optimal to evaluate the results of the ensemble simulations with the large dataset used in the companion paper (Dörenkämper et al., 2020) and with the full downscaling model chain. Nevertheless, the results of the evaluation of the production run with the data included there, support the performance of the configuration selected herein.

*Code availability.* The WRF model code is open source code and can be obtained from the WRF Model User's Page (http://www2.mmm.ucar.edu/wrf/users/, doi:10.5065/D6MK6B4K). For the NEWA production run we used WRF version 3.8.1. The code modifications as well as namelists, tables and domain files we used are available from the NEWA GitHub repository: https://github.com/newa-wind/Mesoscale and permanently indexed in Zenodo (Hahmann et al., 2020). The WRF namelists and tables for all the ensemble members are also available in the repositories. The code used in the calculation of EMD metric is available from: https://pypi.org/project/pyemd/

*Data availability.* The NEWA data is available from https://map.neweuropeanwindatlas.eu/. The forcing data for the mesoscale simulations are publicly available:

ERA5 - https://climate.copernicus.eu/climate-reanalysis,

OSTIA - http://marine.copernicus.eu/services-portfolio/access-to-products/?option=com_csw&view=details&product_id=SST_GLO_SST_L4_NRT_OBSERVATIONS_010_001,

CORINE - https://land.copernicus.eu/pan-european/corine-land-cover,

ESA-CCI - http://cci.esa.int/data.

Some of the tall mast data used for the evaluation of the wind atlas is confidential and thus not publicly available.



**Table A1.** The WRF model setup common to all simulations and to the baseline simulation.

| option | setting |
| --- | --- |
| **Common setup:** | |
| Model grid | D1/D2/D3 with 27 km / 9 km / 3 km horizontal grid spacing |
| | Lambert conformal grid projection |
| Terrain data | Global Multi-resolution Terrain Elevation Data 2010 at 30'' (Danielson and Gesch, 2011) |
| Land use | CORINE land-cover classification (Copernicus Land Monitoring Service, 2019) |
| | ESA-CCI land-cover (Poulter et al., 2015) outside the CORINE domain |
| Vertical discretisation | 61 vertical levels with model top at 50 HPa. |
| Model levels | 20 model levels below 1 km |
| Diffusion | Simple diffusion (option 1), 2D deformation (option 4) |
| | 6th order positive definite numerical diffusion (option 2) |
| | No vertical damping |
| | Positive definite advection of moisture and scalars |
| **Baseline setup:** | |
| Forcing data | ERA5 (Copernicus Climate Change Service (C3S), 2019) reanalysis at 0.3° on pressure levels |
| Sea surface temperature | Operational Sea Surface Temperature and Sea Ice Analysis (OSTIA, Donlon et al., 2012) |
| | fractional sea-ice activated |
| | Lake temperatures from time-averaged ERA5 ground temperatures |
| Cloud micro-physics | WRF Single-Moment 5-class scheme (Hong et al., 2004) |
| Cumulus convection | Kain-Fritsch Scheme (Kain, 2004); D1 and D2 |
| PBL scheme | MYNN level 2.5 (Nakanishi and Niino, 2009) |
| Surface layer scheme | MYNN (Nakanishi and Niino, 2009) with mods (see text) |
| Land surface model | Unified Noah Land Surface Model (Tewari et al., 2004) |
| Shortwave and longwave radiation | RRTMG (Iacono et al., 2008) at 12 minute interval |
| Nesting | one way nesting with smooth (option 2) |
| Nudging | spectral nudging U, V, T and q on D1 |
| | above level 20, no PBL nudging |
| Nudging constant | $0.0003\,\text{s}^{-1}$ |
| Nudging wavelength | 14 (x) and 10 (y) equivalent to $6 \times \Delta x$ of ERA-Interim reanalysis grid spacing |





**Table A2.** Explanation for the various digits of the sensitivity experiments that refers to the code available in GitHub.

| digit | option | convention |
|---|---|---|
| 1 | IC/BC data | E: ERA5, I: ERA-Interim, M: MERRA2, C: CFSR2, F: FNL |
| 2 | Land IC | same as digit 1 (E, I, M, C, F), G: GLDAS |
| 3 | SST data | S: OSTIA, H: HRSST, O: OISST, or same as digit 1 (E, I, M, C, F) |
| 4 | WRF version | 6: WRFV3.6.1, 8: WRFV3.8.1 |
| 5 | Roughness option | 1: constant, 2: annual cycle, 3: aggregated |
| 6 | Separator underscore | |
| 7 | Land Surface Model | Code as in WRF: Thermal diffusion=1, NOAH=2, RUC=3, CLM4=4, PLX=7 |
| 8 | PBL scheme | Code as in WRF: YSU=1, MYJ=2, QNSE=4, MYNN2=5, MYNN3=6, ACM2=7 |
| 9 | Surface layer | Code as in WRF: Revised MM5=1, M-O=2, QMSE=4, MYNN=5, P-X=7 |
| 10 | Modified PBL and surface layer? | no=0, yes=1 |
| 14–15 | Cloud Microphysics | Code as in WRF: WSM5=04, Thompson=08, Thompson+aerosol=28 |
| 16–17 | Convective scheme (D1,D2) | Code as in WRF: No=00, K-F=01, B-M=02, Grell-Devenyi=93 |
| 18–19 | SW/LW radiation | Code as in WRF: CAM=03, RRTMG=04, Fast RRTMG=24 |
| 20–21 | Separator (if need) + extra option | A, B, C, e.g. two-way nesting |

*Author contributions.* AH wrote the first draft, coordinated the sensitivity experiments and analysed some of the results. TS carried out the verification of the model results and worked on the application of EMD metric. All authors participated in the design and conduction of the sensitivity experiments and in the writing and editing of the manuscript.

*Competing interests.* The authors declare no competing interests.



*Acknowledgements.* The European Commission (EC) partly funded NEWA (*NEWA- New European Wind Atlas*) through FP7 (topic FP7-ENERGY.2013.10.1.2) The authors of this paper acknowledge the support the Danish Energy Authority (EUDP 14-II, 64014-0590, Denmark); Federal Ministry for the Economic Affairs and Energy, on the basis of the decision by the German Bundestag (Germany - ref. no. 0325832A/B); Latvijas Zinatnu Akademija (Latvia); Ministerio de Economía y Competitividad (Spain -refs. no. PCIN-2014-017-C07-03,

PCIN-2016-176, PCIN-2014-017-C07-04 and PCIN-2016-009); The Swedish Energy Agency (Sweden); The Scientific and Technological Research Council of Turkey (Turkey-grant number 215M386).

ANH additionally acknowledges the support of the Danish Ministry of Foreign Affairs and administered by the Danida Fellowship Centre under the project "Multiscale and Model-Chain Evaluation of Wind Atlases" (MEWA) and the ForskEL/EUDP (Denmark) project Offshore-Wake (PSO-12521/EUDP 64018-0095).

The tall mast data used for the verification has been kindly provided by the following people and organisations: Cabauw, Data provided by Cabauw Experimental Site for Atmospheric Research (Cesar), maintained by KNMI; FINO 1,2,3, German Federal Maritime And Hydrographic Agency (BSH); Ijmuiden, data from the Meteorological Mast Ijmuiden provided by Energy Research Center of the Netherlands (ECN), processed data shared by Peter Kalverla from Wageningen University, Høvsøre, Østerild, Risø, data provided by Technical University of Denmark (DTU). Most of the WRF model simulations were initialised using ERA5 data, downloaded from ECWMF and Copernicus

Climate Change Service Climate Data Store.

We acknowledge PRACE for awarding us access to MareNostrum at Barcelona Supercomputing Center (BSC), Spain, without which the NEWA simulations would not have been possible. Part of the simulations were performed on the HPC Cluster EDDY at the University of Oldenburg, funded by the German Federal Ministry for Economic Affairs and Energy under grant number 0324005. This work was partially supported by the computing facilities of the Extremadura Research Centre for Advanced Technologies (CETA-CIEMAT), funded by the

European Regional Development Fund (ERDF), CIEMAT and the Government of Spain. In addition, simulations carried out as part of this work also made use of the computing facilities provided by CIEMAT Computer Center.

Caroline Draxl and Gert-Jan Steeneveld are thanked for their earlier review of the WRF model sensitivity experiments. Finally, we would like to thank the project and work package leaders of the NEWA project: Jakob Mann, Jake Badger, Javier Sanz Rodrigo and Julia Gottschall.





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
