# Peer review of "The Making of the New European Wind Atlas, Part 1: Model Sensitivity"

_Geoscientific Model Development, 2019_

## Referee Comment (RC1) · Anonymous Referee #1 · 4 Apr 2020

General comments

This paper describes an impressive set of sensitivity experiments performed with the WRF mesoscale meteorological model, so as to obtain an optimal model configuration for the production of a New European Wind Atlas: a dataset that may well become very influential in shaping Europe's renewable energy landscape. The potential impact of this dataset justifies its documentation in the scientific literature. The vast number of combination of settings that were tested make this paper relevant and interesting for the audience of Geoscientific Model Development.

I should point out that there is significant overlap between this manuscript and an earlier technical report (https://zenodo.org/record/2682604#.XnZH1VHQg5k). Now, while I really support the early reference in anticipation of a definitive journal publication, my

impression is that the manuscript still reads like a technical report. In that respect, it doesn't help that the study is presented as a fait accompli: a mere justification of the NEWA setup that can no longer be changed. Yet whilst the production of the wind atlas has finalized, I think there's actually a lot the authors can still do to make this paper useful for the audience of GMD.

First and foremost, the dataset could (should?) be made publicly available. The data availability section only refers to the final NEWA product, not to the sensitivity experiments upon which the presented results have been based. This is not just a reproducibility issue. Many interesting research and model development questions beyond the scope of NEWA can be addressed with this rich sensitivity dataset, and it would be a waste not to share it.

Furthermore, the discussion is very limited in scope. There is no comparison with similar efforts (although smaller in scope), based on different models. The discussion stays away from any physical interpretation and lacks critical reflection on important choices that have been made. The impact beyond the NEWA project is not considered at all. For example, the authors state that "it would have been optimal to evaluate the results of the ensemble simulations with the large dataset used in the companion paper". But this can still be done, and although the insights would not propagate to NEWA, they could clarify some of the questions that currently remain unanswered. The last of the specific comments lists further issues that I would like to see in the discussion.

Some minor aspects of the model configurations are not documented, which hampers reproducibility. For example the determination of vertical levels or the parameters of the lambert projection. Perhaps the authors could share the namelist of the final configuration? It is also not clear whether the WaSP downscaling methods has been applied to the presented results (and if so, it should be documented).

Specific comments

[Figure]

P2 L14: While it is very clear in the abstract, I miss a sentence like: "This paper describes our efforts to find an optimal configuration of the WRF mesoscale weather model for the production of a New European Wind Atlast (NEWA)." in the introduction. The configuration of WRF for the production of NEWA is the main focus of the paper, yet its introduction is a bit out of the blue with a reference to Petersen 2017. It would be good to provide more context about NEWA. Why was WRF chosen, for example? Given that virtually all options within WRF are investigated in this study, presenting the choice for WRF itself as a an accomplished fact feels a bit unsatisfactory. Line 14 in particular starts with "Given the EWA is 30 years old", which begs for something like "A and B bundled forces to produce an updated wind atlas."

P2 L10: perhaps explain 'the so-called wind atlas method' in one or two sentences? Is this the same method referred to in P2 L22? And is this method also used for the evaluations presented here? P23 L31 makes me think it is indeed, yet P24 L17 seems to suggest the opposite (but it is a bit unclear what is meant by "the full downscaling model chain"). If no further downscaling is used for this study, perhaps don't mention it at all.

P3 L 12-29: this paragraphs seems a bit out of place. I suggest moving it to somewhere around P3 L4-6, such that P3 L30 logically follows after the part about "The approach in NEWA". Perhaps the statement about "best practice setup" can then also be combined with the reference setup referred to in P3 L34.

P4 L20: This requires further discussion, as land surface/soil moisture 'memory' is known to significantly affect the results.

P5 Fig1: All masts seem to be located in the northernmost domains (compare with Fig3). If the configuration was optimized for Northern Europe, what does this mean for the validity of NEWA for the South-European domains?

P6 L8: "Due to difficulty ... has not been filtered or corrected". This requires more justification. At least the authors could say something about how the performance

differs between the various masts or between wind direction sectors. That should provide some intuition about the potential effect of wind farm distortions. It might also be relevant to mention the wind directions that were filtered for FINO, Riso and Hovsore explicitly. Are these prevailing wind directions or not? And how do they relate to the nearby coastlines? Especially in coastal areas, I think it is not safe to assume that model performance is uniform across all wind directions.

P6 L15: While I believe the presented evaluation metrics achieve the stated objective of selecting a single best model configuration for the production of the NEWA (in terms of wind speed), their presentation is quite unclear. I would advise to use the more common term "mean absolute error" (MAE) instead of 'absolute bias'. Also I would advise against making all metrics 'relative', which is mostly confusing. Comparison against a baseline (or: reference) is very good. However, isn't the more common approach to use their fraction rather than the difference? See for example literature on fractional skill score, or the excellent textbook by Wilks (statistical methods in the atmospheric sciences). You would get SS = 1 – (MAE / MAE_ref), and SS = 1 – (EMD / EMD_ref), which would approach 1 for a perfect forecast and 0 for no improvement over the baseline. I suppose that such a uniform scoring system would help to judge whether an improvement in one metric is worthwhile if it is accompanied by deteriorating scores for other metrics or locations. Right now, that's not clear (see e.g. my specific comment P18 L5).

P6 L19: "The main goal of the NEWA project was the evaluation of the wind climate, which is usually understood as the probability distribution of wind speed and direction at a specific point". Why then, is wind direction not evaluated at all in this manuscript? And what about vertical wind shear?

P6 L24: This statement is quite irrelevant and I doubt if it's always true. I suggest to remove it.

P6 L29: Move part about RMSE to after the stuff about bias. Also, perhaps refer to a

paper about skill-scores. Part about comparing to baseline/reference setup is a good idea and might be useful for others that want to learn from this study. Therefore, a very clear explanation is appropriate. I had to read it three times.

P7 Fig2: I understand that the histogram representation of the wind speed distribution is appealing because it is widely known. Panel A succeeds in showing the difference between EMD and absolute bias, but I wonder if this cumulative distribution plot would be even more intuitive. Also, I'm curious why the difference between EMD and absolute bias is larger for small absolute bias.

P7 L15: The EMD explained as the are between CDFs is very intuitive. It took some effort to verify this, but eventually I found it (https://stats.stackexchange.com/a/299391). It seems that this statement is only true for univariate distributions. A reference here would be appropriate.

P8 L16: I understand that the authors try to put emphasis on the differences (or rather: the absence thereof) between the geographical domains, especially seeing that PBL is further investigated later on. It is indeed a good idea to test this domain-sensitivity with various set-ups. But the section is written such, that the reader tends to focus mostly on the performance between PBL schemes rather than geographical domain. This is especially true towards the end of the section, where it seems that conclusions are drawn about the reference configuration, rather than about the domains. Both figures 5 and 6 contribute to this shift of focus.

P8 L19: I think it would be good to briefly explain the differences between these two PBL schemes, and why these two schemes were chosen. PS: or in the later section.

P9 Fig3: The experimental sites don't seem to correspond to the locations of the masts used for the evaluation presented in this paper. What then, is the reason to show these sites? Perhaps this figure could be merged with Figure 2? Also, the abbreviation "PD" is not clear to me.

P9 Tab2: It would help the reader if the acronyms (particularly the meaning of S1 and W1) was explained in the text/caption.

P8 L26: "the largest differences arise from the choice of PBL scheme, as shown in Fig 4". While the figure clearly illustrates the point that the authors make about the coincidence of regions with high surface roughness with areas of large differences between PBL schemes, it does not actually show, as the authors claim, that this is the largest difference. But even if it's not the largest difference, it would still be interesting to also show/quantify the effect of the different initialization strategy. Moreover, in the light of the authors' excellent point about the necessity to quantify differences between distributions, I'm quite surprised that they opted here to show the difference in the mean annual wind speed, rather than the more comprehensive EMD.

P11 L4: I'm a bit concerned about the authors' conclusion that the weakly nudged setup is actually the best choice. Particularly, I would like to see whether the evaluation statistics are dependent on the lead time of the simulation.

P11 L9: Change title? Most of the section is about the modifications to MYNN.

P12 Fig6: Is this figure for all mast heights? And are the differences shown here actually significant? Especially the correlation seems very consistent between all runs. And what about the earth mover's distance? Why is it not shown here? Is the bar plot really the best choice here, seeing that differences are amplified or dampened depending on the choice of the axes' intersection?

P13 L5: It would be useful to describe how these 25 configurations where selected from on the thousands of combinations alluded to before. Perhaps repeat or elaborate on the 'expert judgment' here.

P16 L16: "absolute difference in relative bias". This formulation is incorrect. A correct formulation would be "Fig 9b shows the difference in absolute relative bias between ...".

P17 Fig9: I would suggest to group figures 9 and 10 together, OR, to present 9a and

10a together, and 9b and 10b. As it is now, it is difficult to compare figures 9b and 10b. Also, consider using a different colormap for a and b, since right now green means "good" in b, but not in a, in both figures.

P18 Fig10: It is not clear to me how the 'relative' EMD is calculated in panel A. And is the same 'relative' EMD used for panel B? Why not just show the EMD in m/s? I feel the author's are making things needlessly complicated. Same question applies to the 'absolute bias'. Although I can see that the 'non-relative' metrics are wind-speed dependent, mean wind speeds are all around 10 m/s, so the differences between sites will be very small. Therefore I would argue: simpler is better.

P18 L5: I'm not sure if the choice for MO is justified based on the statistics shown. Although the EMD improves slightly for four sites, it degrades severely for some of the others. I'm not sure of the overall effect is positive. This could use some extra discussion.

P19 L8: This is interesting indeed. Perhaps the authors can discuss this observation a bit more in depth? I'm still not convinced that 8-day nudged simulations are the best choice.

P20 Fig11: Same comments as for fig 9 and 10: it would be better to use a different colormap for figures a and b, and perhaps group all figures together to prevent them spreading over multiple pages. Also reconsider using relative/normalized metrics.

P20 L6: "at hub height". Does this mean that only $\sim$100 m was used for all tables? So far I wasn't sure, but I was under the impression that the metrics were calculated on the basis of all measurement heights. What does this mean for the representation of the (distribution of) wind shear between the various model simulations? I know that the mean profiles have very similar shear, but beware that instantaneous profiles can show substantial variation!

P20 L7: "Unfortunately, we did not run ... so we cannot". This statement contributes

substantially to the overall impression that this manuscript is an accomplished fact.

P21 L3: It seems a bit weird that this is the last experiment. If I would have designed this experiment, it would have been the first, as the other settings may depend on it. Especially the combination of domain size and nudging/initialization strategy seems influential.

P21 L8: This is an interesting dilemma. Did the authors modify the WRF registry to output only the relevant parameters? Would the 'restart' option not lift this constraint as the simulation time could be shortened to enable intermediate postprocessing? And how does the pan-European domain compare to the CONUS domain used in the rapid refresh configuration of NOAA? Have the authors contacted them for advise about their reference setup and HPC strategy? Options to stream the WRF output, or to access model fields during a simulation to postprocess them right away would be very welcome recommendations for model development. I wonder whether such features are already available, for example through the 'basic model interface' developed by CSDMS.

P21 L11: "outside region of interest . . . would be wasted". I have to disagree here. Although it is not the explicit goal of the NEWA project, these data could be very useful for those non-EU countries. Again, please broaden the scope from "NEWA" to "a relevant and interesting dataset for the audience of GMD". I think this dataset can have more impact if it would be available for other researchers as well. The term 'waste' therefore rubs me the wrong way.

P21 L26: (and possibly . . . not shown). Model runs that would show this have also not been described as far as I can see. What additional simulations did the authors perform that inspire this statement, or is it mere speculation?

P21 L34: "We decided, however, against very small domains. In terms of accuracy they would probably perform better". Not only does this statement sound speculative, it also partly undermines the objective of the paper. If one of the options considered (the SM domains) was not an option to begin with, why test it? For some sites, the impact of this

Interactive
comment

decision seems to be larger than the accuracy gained through the detailed optimization of all other settings of the model...

P22 Fig13: The y-axis is unreadable.

P24 L2-3: "In that paper we conclude that..." ? Better than just using 'raw' ERA5 data?

P24 L4: "some questions remain unresolved ... expensive nature of the numerical experiments". This is obviously true, but I feel there are many more questions unanswered because of limited manpower. I'd really appreciate it if the authors could reflect more on that aspect of their study.

P24 L17: "It would have been optimal..." again this contributes to the "accomplished fact" feeling. This can still be done, can't it? And it can answer some of the questions I have asked, e.g. P5 Fig1 related to the representativeness of the northern domains for Southern Europe.

P24: The discussion (or other parts of the paper if appropriate) should also address why vertical resolution was not subject to sensitivity analysis, what the uncertainty of the observations is, why wind direction is not considered at all, whether performance is similar across different heights, why/how wind shear has (not) been assessed, how the set-up compares to other similar efforts. The outlook should offer some advice for future studies: what have we learned from this study, in what direction should model development evolve, what are the main strengths/weaknesses of the WRF setup, which parameterization schemes should we abandon right away, etc.

Technical corrections

Excessive use of commas and conjunctions make parts of the text difficult to read. This can easily be addressed by making shorter sentences. For example: - P2 L6-7 rewrite "but not only" - P2 L7-8 use only on of "for example ... to name a few" - P2 L13-15 suggest " ... its usefulness. It has ..." - P3 L13-16 start new sentence at "however" - P3 L10 start new sentence at "however" - P3 L16-18 move "has been reported" to

beginning: "a number of studies report ... - P3 L19 remove comma after "cases", suggest: "two processes with opposing effects" (remove "canceling each other out") - P7 L12 suggest: "Small changes in wind speed are (thus) amplified when converted to power." - P8 L18-20: suggest to split in 2 or 3 shorter sentences. Remove "the aim was", as the next sentence also states "the objective". - P8 L22: "or if there were regional differences" can be omitted as it is already implied by the use of "whether" - P10 L4: better to split up and rephrase, instead of using "but" twice in the same sentence. - P10 L6: this sentence can also be split in two shorter sentences. - P11 L6: Unclear, long sentence. - P22 L8: unclear sentence; a.g.l. and AGL are the same. Which figures? - P23 L18: weird use of commas around "... change source..." - P24 L6-7: "however ... but ..."

Other editorial remarks: • P3 L12: "A large number" or "Large numbers of" • P3 L21: citation without brackets • P3 L28: coastal winds? Flow is ambiguous (air or water). • P4 L8: Simulations (plural), or perhaps "reference configuration"? • P6 L23: remove "in" • P 8 L20: remove "left" (or write "left untouched"?) • P15 L2: "regional" should be "region(s)"? • P17 L20: "conclusions can be drawn" • P18 L1: "scheme and run" both refer to a scheme/set-up/configuration, right? • P21 L21: "six" instead of 6 (in line with the surrounding text) • P21 L34: rephrase "which would face" • P24 L2: "wind climate" • P24 L10: "best optimal" • P24 L17: "observational dataset"

---

## Referee Comment (RC2) · Anonymous Referee #2 · 17 Apr 2020

General Comments

The paper summarizes an exhaustive sensitivity analysis performed to inform the final model setup of the New European Wind Atlas. This surely must be the most extensive such analysis to date and overall is an impressive achievement. The novel use of the Earth Mover's Distance is also applauded and clearly offers a much-needed complimentary metric alongside the typical timeseries-based performance metrics.

I believe this paper should ultimately be published; however, I have several comments and concerns about the work that have not been addressed in the paper. First, all of the critical validation was performed in Northern Europe, despite the NEWA being produced for Europe and Turkey as a whole. I realize that computational expense and data availability/quality were probably a factor, I can't help but feel that with such collaboration across European institutes that a more regionally diverse validation campaign could have been performed. Of course NEWA has already been produced, but I think some critical commentary on how validation in Northern Europe (with its unique climatology) would apply across other climates in Europe with their own unique climatologies is needed here. Otherwise, the paper reads as if the idea of more extensive validation was overlooked.

Furthermore, I did not find sufficient presentation of results to justify selection of the final model setup. Rather, a wind profile plot and two heat maps of bias and EMD were provided, and it seemed very quickly the section was wrapped up with the final model selection. I think some further synthesis is required, such as a table of figure showing mean bias, RMSE, EMD, etc. across all validation sites. Without this, in my opinion, the selection of the final model setup seems unjustified.

Finally, as far as I can tell, ERA-interim was used in the sensitivity analysis, but ERA-5 was used in the final production run. This point is not discussed in this paper but I think it's an important one. Does existing research suggest bias or EMD differences between the two data sets? If so, what are the implications on selecting the best model setup using one large-scale forcing but pivoting to a new product for the actual production runs?

In conclusion, I think this is a valuable contribution to the literature. However, several key limitations of this study need to be sufficiently addressed and discussed before final publication. In addition, a couple summary figures and tables would help justify final model selection.

Specific Comments

Page 1, Line 9: Why were sensitivity experiments only conducted in Northern Europe when the data set was for Europe as a whole? Surely tall masts must be available elsewhere? If this was a decision based on computational restrictions, this should be stated and the implications of this smaller validation domain, in the context of regional

wind climates, should be discussed.

Page 2, Line 15: Can 'linearized model' be described more, or at least a couple references listed to provide background?

Figure 1: As in comment in Line 9, validation only in Northern Europe poses a problem for a product that covers Europe as a whole. This key study limitation needs to be discussed in detail.

Table 1: What is the time resolution of the observed data used to indicate sample size? I'd assume hourly but please make this clear.

Page 6, Line 9: Given the known impact of turbine wakes at these measurement sites, why not filter the data by wind direction to ensure the data are free stream? Especially in such a detailed sensitivity analysis where performance metrics between different model setups can be on the order of 0.1 m/s, allowing wakes to affect the measurement data seems inappropriate.

Page 7, Line 14: I'd use 'interpreted' rather than 'understood' when describing EMD as a measure of physical work.

Page 7, Line 15: Given the novelty of the EMD metric, I wonder if a new Figure showing the area between cumulative distribution functions would be useful, given this is how the metric is actually computed.

Page 7, Line 16: What are circular variables and why are they relevant here? Are you validating wind direction?

Page 8, Line 23: Why was WRF 3.6.1 used, given it is 6 years old and the significant advances made since then? Was this part of an older study that is now being published?

Page 10, Line 3: But MYNN winds are higher in the NW offshore domain and lower in the SW domain. Can you discuss? Is NW offshore domain generally more stable?

Figure 6: Given the detailed justification of EMD earlier, why is it not being used here?

Figure 8: I'm struggling trying to distinguish the different model runs. Multiple setups seem to have identical markers (at least to the naked eye). Also the lines are so tightly clustered that it's generally not possible to discern one profile from another. As such the Figure does not provide much useful information and I would recommend revising or deleting.

Figure 9a: Would an additional column showing average across sites be useful in identifying the best performing model setup?

Figure 9b: I'm not sure I see the value of performance metrics relative to the 'base' setup. In my mind this base setup is just another member of the ensemble and not otherwise special. So why compare all ensembles against this one? Do we know it to be the most accurate? If not, I don't see the value in this relative comparison. Please justify.

Figure 10b: Likewise to comment above. I'm not seeing the value of this relative comparison.

Page 18, Line 5: This is a big jump to conclude the best performing model setup based on the figures shown in this section. For example, the improved performance of MO over the Base and MM5 setups isn't clear from the profile plots or the heat maps. I think some final figure or table is needed showing key performance metrics averaged across all sites in order to justify this model choice.

It also seems that the multi-physics sensitivity analyses and the selection of final production run in Section 5.3 was done using ERA-interim as the large scale forcing in WRF. However, ERA-5 was used in the final NEWA. This seems problematic given potential differences (e.g., biases) between the two data sets. I understand that ERA-5 was not available at the time these simulations were performed; however, some discussion around the implications of changing the large scale forcing without sensitivity

analysis needs to be provided.

Page 19, Line 8: Unclear how ERA5 reanalysis slow down of winds relates to a sensitivity analysis of ERA-interim, FNL, and MERRA2. Was ERA5 part of this comparison?

Figure 11a and 12a: What is the difference between BASE and ERAI? I thought the base run was done using ERA-interim.

Figure 11b and 12b: Same comment as previous.

---

## Author Comment (AC1) · 26 Jun 2020

**Response to Referee #1**

Thank you for the comprehensive comments, and also for taking the time to truly read through our manuscript. We feel that your comments were very helpful for increasing the quality of the paper to its current level. Your comments, together with those of referee #2, led to a thorough revision of the paper.

The most general comments regarding the revisions to the manuscript are:

1. At the start of the research project typically there are high expectations placed on the sensitivity experiments, however, reality always brings some corrections and caveats. Given the enormous possibilities in setting up WRF, an "optimal" configuration is unreachable. We have tried to revise the introduction to convey that the paper focuses on finding the "best possible" model configuration **constrained** by the practical issues in running the model simulations and the ultimate goal to use the simulations for a **wind atlas**.

2. The manuscript aims to tell the story of how the NEWA wind atlas came to be. Therefore, further analysis of the model results will make the flow of the paper less clear. We have tried to enhance this structure in the revised manuscript.

3. We have replaced some of the figures (6, 9–12, 13) to homogenise the analysis of the results. We have also added new figures including the RMSE and circular EMD for wind direction.

4. We strengthened the connection to the companion paper, https://www.geosci-model-dev-discuss.net/gmd-2020-23/, which is now available.

The reviewers' comments are in black and our responses in blue.

**General comments**

1. First and foremost, the dataset could (should?) be made publicly available. The data availability section only refers to the final NEWA product, not to the sensitivity experiments upon which the presented results have been based. This is not just a reproducibility issue. Many interesting research and model development questions beyond the scope of NEWA can be addressed with this rich sensitivity dataset, and it would be a waste not to share it.

   We agree. However, the subset of the WRF model data from the simulations totals 15 TB. We are trying to find a solution, perhaps via an EUDAT grant. As a minimum we will make available the yearly wind statistics from each simulation in Zenodo.

2. Furthermore, the discussion is very limited in scope. There is no comparison with similar efforts (although smaller in scope), based on different models. The discussion stays away from any physical interpretation and lacks critical reflection

on important choices that have been made. The impact beyond the NEWA project is not considered at all. For example, the authors state that "it would have been optimal to evaluate the results of the ensemble simulations with the large dataset used in the companion paper". But this can still be done, and although the insights would not propagate to NEWA, they could clarify some of the questions that currently remain unanswered. The last of the specific comments lists further issues that I would like to see in the discussion.

We agree with the statements above, however we feel that the narrow nature of the manuscript is justified taking into account that there exists a second part to this study in a companion manuscript that is dedicated to critically evaluating the results of the final choice. Also, as mentioned above, the manuscript tries to tell a coherent story of the choices made during the creation of NEWA wind atlas. Including further data analysis at this stage might just increase the confusion. Also, we believe the manuscript is already long enough. We believe the scope of the manuscript is still useful to modellers both in wind energy and in wider applications of numerical weather prediction models.

Even if we wished to do further evaluation with the data used in part 2, it would not be possible. Figure 4 in part 2 shows the tall masts available in the Vestas database. The comparison will be limited since there are no masts in Denmark and very few in Germany. In Poland data exists in some masts but for only a few months in 2015.

3. Some minor aspects of the model configurations are not documented, which hampers reproducibility. For example the determination of vertical levels or the parameters of the lambert projection. Perhaps the authors could share the namelist of the final configuration? It is also not clear whether the WaSP downscaling methods has been applied to the presented results (and if so, it should be documented).

All namelists are shared in the project GitHub (https://github.com/newa-wind) and in the NEWA Zenodo (http://doi.org/10.5281/zenodo.3709088 site, this also includes all the "geo" files used in the model simulations. The link to zenodo is located under "assets" in the manuscript GMD website. Therefore, the simulations are reproducible. The full description of the NEWA model grid is in the companion manuscript [1].

As for the second question. No further downscaling is done in this Part 1. All the evaluations against tall masts are done with the raw WRF model data. The sites are relatively simple (offshore and over flat terrain), where the microscale modelling will add little to the mesoscale model solution.

**Specific issues**

1. P2 L14: While it is very clear in the abstract, I miss a sentence like: "This paper describes our efforts to find an optimal configuration of the WRF mesoscale weather model for the production of a New European Wind Atlas (NEWA)." in the introduction. The configuration of WRF for the production of NEWA is the main focus of the paper, yet its introduction is a bit out of the blue with a reference to Petersen 2017. It would be good to provide more context about NEWA. Why was WRF chosen, for example? Given that virtually all options within WRF are investigated in this study, presenting the choice for WRF itself as a an accomplished fact feels a bit unsatisfactory. Line 14 in particular starts with "Given the EWA is 30 years old", which begs for something like "A and B bundled forces to produce an updated wind atlas."

   Agreed. We have added a new paragraph about the wider NEWA project in the introduction. The rationale for using the WRF model is also included.

2. P2 L10: perhaps explain "the so-called wind atlas method" in one or two sentences? Is this the same method referred to in P2 L22? And is this method also used for the evaluations presented here? P23 L31 makes me think it is indeed, yet P24 L17 seems to suggest the opposite (but it is a bit unclear what is meant by "the full downscaling model chain"). If no further downscaling is used for this study, perhaps don't mention it at all.

   The wind atlas method is mentioned in the introduction because it was used to create the earlier European Wind Atlas. But we understand that can be confusing and reference to it was removed from the rest of the paper.

3. P3 L 12-29: this paragraphs seems a bit out of place. I suggest moving it to somewhere around P3 L4-6, such that P3 L30 logically follows after the part about "The approach in NEWA". Perhaps the statement about "best practice setup" can then also be combined with the reference setup referred to in P3 L34.

   Excellent suggestion, thanks. The three paragraphs starting in P2, L25 and ending in P3, L29 have now been restructured in a more logical way: (1) adapting models to wind energy applications, (2) review of previous ensemble studies, and (3) approach taken in this paper.

4. P4 L20: This requires further discussion, as land surface/soil moisture "memory" is known to significantly affect the results.

   We did consider the issue of land surface and soil moisture memory. In regional climate model simulations, the NWP model parameterisation are often tuned to avoid model drift (e.g. [3]). For generating a wind atlas, we are not worried about model drift because the simulations are re-initialised often (here - every 7 days). It would be optimal if we could re-initialise the atmospheric model often, but keep the state of the land surface from one simulation to the next. However,

for practical reasons this is not possible, since the simulations had to be run sequentially. Also, it is not obvious that the precipitation produced by the WRF model is accurate enough to keep the soil moisture from drifting. Lastly, the land surface and soil moisture memory is indeed important in simulating climate-relevant parameters such as temperature and precipitation, but we don't think there is enough evidence that it is critical in wind reanalysis. The connection between soil moisture, sensible heat flux, wind profile and wind speed does exist, but we do not have a systematic way of validating it in the context of NEWA. We have tried simulations initialised with Global Land Data Assimilation System (GLDAS) data, but the results were inconclusive.

5. P5 Fig1: All masts seem to be located in the northernmost domains (compare with Fig3). If the configuration was optimized for Northern Europe, what does this mean for the validity of NEWA for the South-European domains?

Yes, unfortunately there are very few quality tall (above 50 m height) masts in Europe publicly available. At the start of the project, we had eight sites in Northern Europe and another hand-full in the other domains. It is not optimal, but we had no other option. Winds observations from surface stations are plentiful, but often they are placed in complex sites that make the evaluation difficult, and the accuracy of a model at 10-m height does not give any warranty that the WRF configuration will also be accurate at wind turbine height [2]. In this manuscript we argue that the combinations of PBL/SL parameterisation schemes behave the same way in different regions in Europe (section 5.1) and thus conclusions from Northern Europe are also applicable to the other regions. We believe that this statement and approach is supported by the evaluation results of the wind atlas that are described in companion manuscript (Part 2)

6. P6 L8: "Due to difficulty . . . has not been filtered or corrected". This requires more justification. At least the authors could say something about how the performance differs between the various masts or between wind direction sectors. That should provide some intuition about the potential effect of wind farm distortions. It might also be relevant to mention the wind directions that were filtered for FINO, Riso and Hovsore explicitly. Are these prevailing wind directions or not? And how do they relate to the nearby coastlines? Especially in coastal areas, I think it is not safe to assume that model performance is uniform across all wind directions.

The wind sector filtered for mast distortion is now explicitly listed in Table 1. We have also corrected the height of FINO1 and FINO2 and added the height of the wind direction data used in the analysis.

7. P6 L15: While I believe the presented evaluation metrics achieve the stated objective of selecting a single best model configuration for the production of the NEWA (in terms of wind speed), their presentation is quite unclear. I would advise to

use the more common term "mean absolute error" (MAE) instead of "absolute bias". Also I would advise against making all metrics "relative", which is mostly confusing. Comparison against a baseline (or: reference) is very good. However, isn't the more common approach to use their fraction rather than the difference? See for example literature on fractional skill score, or the excellent textbook by Wilks (statistical methods in the atmospheric sciences). You would get SS = 1 - (MAE / MAE_ref), and SS = 1 - (EMD / EMD_ref), which would approach 1 for a perfect forecast and 0 for no improvement over the baseline. I suppose that such a uniform scoring system would help to judge whether an improvement in one metric is worthwhile if it is accompanied by deteriorating scores for other metrics or locations. Right now, that"s not clear (see e.g. my specific comment P18 L5).

*That is a very good suggestion, thanks. We have now revised Figures 9–12 to use this "skill score (SS)" as $SS = 1. - (MM/MM_{ref})$, where $MM$ is BIAS, RMSE, EMD or CEMD.*

8. P6 L19: "The main goal of the NEWA project was the evaluation of the wind climate, which is usually understood as the probability distribution of wind speed and direction at a specific point". Why then, is wind direction not evaluated at all in this manuscript? And what about vertical wind shear?

   *That is a very good point. We have now included the evaluation of the wind direction for the initial simulations and the large ensemble. However, as mentioned above, the manuscript tells a story of how we arrived to the final NEWA configuration. Adding new parameters such as wind shear, while interesting and relevant, deviate from the main story of the document.*

9. P6 L24: This statement is quite irrelevant and I doubt if it's always true. I suggest to remove it.

   *Agreed. The two sentences have been removed.*

10. P6 L29: Move part about RMSE to after the stuff about bias. Also, perhaps refer to a paper about skill-scores. Part about comparing to baseline/reference setup is a good idea and might be useful for others that want to learn from this study. Therefore, a very clear explanation is appropriate. I had to read it three times.

    *Agreed. The section has been rewritten to objectively describe of the methods used. We also added the new metric for the wind direction, the circular EMD, or CEMD.*

11. P7 Fig2: I understand that the histogram representation of the wind speed distribution is appealing because it is widely known. Panel A succeeds in showing the difference between EMD and absolute bias, but I wonder if this cumulative distribution plot would be even more intuitive. Also, I'm curious why the difference between EMD and absolute bias is larger for small absolute bias.

An additional panel showing the EMD as the area between the cumulative distributions has been added to Figure 2.

The question "why the difference between EMD and absolute bias is larger for small absolute bias" could be reformulated as "why is the difference between EMD and absolute bias smaller for larger absolute bias". If two distributions have the same mean, then the bias is not able to distinguish between them but the EMD can be used to measure how similar are the distributions. If two distributions have the same shape but different means, then the minimal transport necessary to "move" the distributions towards each other will be equivalent to the difference in means. A nice illustrative example of EMD properties can be found in: Lupu et al.[5]. We have added this reference to the article's text.

12. P7 L15: The EMD explained as the are between CDFs is very intuitive. It took some effort to verify this, but eventually I found it (https://stats.stackexchange.com/a/299391). It seems that this statement is only true for univariate distributions. A reference here would be appropriate.

Agreed. The reference Rabin et al. [6] extends the CDF interpretation of EMD to circular variables. The text has been updated to clarify this and the fact that this applies only to one-dimensional distributions.

13. P8 L16: I understand that the authors try to put emphasis on the differences (or rather: the absence thereof) between the geographical domains, especially seeing that PBL is further investigated later on. It is indeed a good idea to test this domain-sensitivity with various set-ups. But the section is written such, that the reader tends to focus mostly on the performance between PBL schemes rather than geographical domain. This is especially true towards the end of the section, where it seems that conclusions are drawn about the reference configuration, rather than about the domains. Both figures 5 and 6 contribute to this shift of focus.

Agreed. The main focus of the section was to show similar sensitivity in various regions and not on the evaluation. We suggest a new structure where section 5.1 relates to the five domains only, and a new section is added where we discuss the validation against the sites in only one of these domains.

14. P8 L19: I think it would be good to briefly explain the differences between these two PBL schemes, and why these two schemes were chosen. PS: or in the later section.

Good point. A sentence has been added to the revised manuscript.

15. P9 Fig3: The experimental sites don't seem to correspond to the locations of the masts used for the evaluation presented in this paper. What then, is the reason to show these sites? Perhaps this figure could be merged with Figure 2? Also, the abbreviation "PD" is not clear to me.

We removed the NEWA experimental sites from the figure. They are not relevant to this paper because data from the experiments were not used in the model validation. "PD" stands for Perdigão, the NEWA experimental site in Portugal. We renamed it "PO", Portugal, to avoid confusion.

16. P9 Tab2: It would help the reader if the acronyms (particularly the meaning of S1 and W1) was explained in the text/caption.

    Agreed. A short explanation has been added.

17. P8 L26: "the largest differences arise from the choice of PBL scheme, as shown in Fig 4". While the figure clearly illustrates the point that the authors make about the coincidence of regions with high surface roughness with areas of large differences between PBL schemes, it does not actually show, as the authors claim, that this is the largest difference. But even if it's not the largest difference, it would still be interesting to also show/quantify the effect of the different initialization strategy. Moreover, in the light of the authors' excellent point about the necessity to quantify differences between distributions, I'm quite surprised that they opted here to show the difference in the mean annual wind speed, rather than the more comprehensive EMD.

    Agreed. The figure does not show that the largest differences arise from the choice of PBL scheme. But, the following figures do. Except for the northwest of the NW domain and mountainous areas in the Pyrenees and the western Alps, the differences are larger in the MYNN-YSU than the W1-S1 comparison. We have toned down the statement in the manuscript.

    Instead of including more maps, which were not used in the original work, Figure 6 in the manuscript now shows the various statistics for the sites, including EMD. At the time this analysis was done we had not yet discovered the advantages of using the EMD.

18. P11 L4: I'm a bit concerned about the authors' conclusion that the weakly nudged setup is actually the best choice. Particularly, I would like to see whether the evaluation statistics are dependent on the lead time of the simulation.

    The plots depicting error metrics as a function of lead time are depicted in Figure 2. The lead times were aggregated into 12-hour bins, and the weighted average over all the stations is shown, with weights being the number of samples available for each bin in each station. The metrics shown are BIAS, the absolute value of BIAS, with the absolute value being taken before the averaging process, RMSE and EMD. No specific pattern can be observed that would describe how the error metrics evolve over time — probably the number of samples is too small and random errors dominate the distribution. On average, the error metrics for weekly runs are smaller than for the daily runs, which confirms the results described in the paper. The EMD for weekly runs is about the same as for the daily runs for MYNN PBL scheme, but YSU weekly runs seem to be associated with slight

[Figure]

Figure 1: Differences in annual mean wind speed at 100 m between pairs of model simulations: MYNN-YSU on the left, W1-S1 on the right for three of the model domains. The colour bar is identical for all panels.

increase in EMD. However, one must take into account the difference in number of samples in each distribution, and it is likely, that EMD penalises the inhomogeneity that arises from the smaller number of samples. The figure shows a slight downward trend for both the BIAS and absolute value of BIAS, consistent with the hypothesis that the increased performance of weekly runs comes from the fact that the model solution is allowed to fully develop the mesoscale circulations,

however, more detailed investigation of this matter is beyond the scope of this paper. The effect of the spin-up time was previously studied in Hahmann et al (2015) [4] and Vincent and Hahmann (2015) [7].

In conclusion, based on the statistics presented in this paper and previous studies, we believe the choice of the weekly simulations is justified. From the answer to Referee #1 (item number 12 in P8,L16) section 5.1 is now split into two sections. In the second of these, we will add a couple of sentences justifying the use of the weekly setup.

[Figure]

Figure 2: Error metrics as a function of lead time. Lead time is aggregated in 12-hour bins. Weighted average over all stations used in the analysis is shown with weights being the number of samples. Please note that the number of samples in each lead time bin for YSUL61S1 and MYNL61S1 runs is much larger than for YSUL61W1 and MYNL61W1, i.e. $\sim 4000$ samples for S1 run bins and $\sim 600$ samples for W1 bins.

19. P11 L9: Change title? Most of the section is about the modifications to MYNN.

    The title of the section heading has now been changed to "Sensitivity to properties of the MYNN scheme"

20. P12 Fig6: Is this figure for all mast heights? And are the differences shown here

actually significant? Especially the correlation seems very consistent between all runs. And what about the earth mover's distance? Why is it not shown here? Is the bar plot really the best choice here, seeing that differences are amplified or dampened depending on the choice of the axes' intersection?

No, the figure shows the metrics for the underlined heights in Table 1. We chose to validate for the height that is closest to the common turbine height $\sim 100\,\mathrm{m}$. This info has now been added to the figure caption.

This figure has been replaced by a new figure that now includes the EMD and CEMD in addition to the BIAS and RMSE. This homogenises the results with the rest of the paper. However, at the time this analysis was done we had not yet discovered the advantages of using the EMD and the choice of model configuration was not based on this measure.

21. P13 L5: It would be useful to describe how these 25 configurations where selected from on the thousands of combinations alluded to before. Perhaps repeat or elaborate on the "expert judgement" here.

For PBL/SL/LSM parameterisation the number of options is more finite because some combinations are technically not possible. At the beginning of our study, the table of experiments contained many more combinations. As mentioned in the following paragraph, many of the combinations simply did not run despite our attempts to do so and were excluded form the final set of experiments.

22. P16 L16: "absolute difference in relative bias". This formulation is incorrect. A correct formulation would be "Fig 9b shows the difference in absolute relative bias between ...".

Agreed. Please see answer to item 7.

23. P17 Fig9: I would suggest to group figures 9 and 10 together, OR, to present 9a and 10a together, and 9b and 10b. As it is now, it is difficult to compare figures 9b and 10b. Also, consider using a different colormap for a and b, since right now green means "good" in b, but not in a, in both figures.

Thank you. This is a good suggestion. We have also homogenised all the figures that rely on "heatmaps". Now any purple values are "good" and any brown values are "bad". The new colour table is colour blind friendly.

24. P18 Fig10: It is not clear to me how the "relative" EMD is calculated in panel A. And is the same "relative" EMD used for panel B? Why not just show the EMD in m/s? I feel the author"s are making things needlessly complicated. Same question applies to the "absolute bias". Although I can see that the "non-relative" metrics are wind-speed dependent, mean wind speeds are all around 10 m/s, so the differences between sites will be very small. Therefore I would argue: simpler is better.

In total agreement. Please see response above (item 23).

25. P18 L5: I'm not sure if the choice for MO is justified based on the statistics shown. Although the EMD improves slightly for four sites, it degrades severely for some of the others. I'm not sure of the overall effect is positive. This could use some extra discussion.

Agreed. This topic deserves a longer explanation

We can start explaining what we mean by the best model setup. The best model setup would have the best verification metrics for all the stations analysed. The problem is that from the sensitivity analysis results it is clear that some model setups have better scores for one verification metric (e.g. EMD) and some setups have better scores in other metrics (e.g. RMSE). Also, the performance of the setup varies considerably from station to station. In addition, one can clearly see that performance at each station is systematic. There are "good" stations where all setups perform well (e.g. FINO1) and "bad" stations where all setups struggle (e.g. Risø). Should the model setup performance count more in the good stations? Or in the bad stations? The model bad performance is often associated with details of the station siting, which are poorly represented in the WRF model resolution used.

In addition, our goal is to choose the model setup that would perform best over the full NEWA domain. Basing this choice on 8 stations introduces significant uncertainty. Taking into account all the results available to us (not all of them included in the manuscript due to the issues of space) and the limitations described above the conclusion is, that there is no single setup that could be easily identified as "the best". Instead, we have a small set of setups, where each performs equally good (or bad) depending on the metric or station. An argument could be made that each of them should be used for the final product. While working on the project, we made the conscious decision not to delegate the decision to a simple algebra of taking the average metrics over all the stations, because that would introduce the assumption that this decision would not change, if for example another station is added.

We would argue that we cannot distinguish the performance of these "good" members based on the observational data available. In addition, due to the limited computational resources available we had to look at the computation aspects of the setups. The MO setup is one of the best performing, according to the verification results. It performs well both in terms of distribution (EMD) and in time-series (RMSE). It has the additional benefit of runs being numerically stable, when compared to BASE, i.e., the runs failed less (this aspect was important, because of the necessity for a person to monitor and re-submit the runs), and MO also had a favourably small computational time when compared to other setups.

We are aware that arguments could be made that some other setup should have been chosen. We would have liked to be able to make this decision using more observations or after some additional analysis, however, due to the practical constrains, a decision had to be made based on what can only be described as "imperfect information", a situation that might be familiar to many readers of this manuscript. Therefore, we would like to argue that the our choice is validated by the good evaluation results described in Part 2. We would like argue that although we are not claiming that we made the "perfect" choice, that is, it is possible that choosing some other setup would have yielded even better metrics, but our choice was "good enough" given the information we had available at the time.

In conclusion. We plan to include some of the points raised above in the revised manuscript especially in the "discussion and outlook" section.

26. P19 L8: This is interesting indeed. Perhaps the authors can discuss this observation a bit more in depth? I'm still not convinced that 8-day nudged simulations are the best choice.

    Agreed. We have expanded the figures to include the RMSE and CEMD. To follow this the discussion, we have expanded the discussion regarding the differences between statistics that look at the distribution versus time synchronisation.

    The new figures added show better agreement not only for the overall distribution (EMD), but also in the RMSE. The stronger nudging towards the reanalysis, which has observational data assimilated in it, in experiments NUDD3 and GNUD3 results in smaller RMSE. The interesting question is: why does it degrade the BIAS and EMD performance? A simple answer is that the WRF model mostly has increased performance over ERA5 (see Part 2 [1]) and therefore nudging towards the poorer performing model decreases model performance. A comprehensive answer to this question is beyond the scope of this paper, however, the conclusion is less surprising than it seems, taking into account the many different interactions between wind speed, surface energy budget, transport in the surface/planetary boundary layer, etc. Nudging is "artificial" or non-physical and can interfere with these complicated processes in non-trivial way.

27. P20 Fig11: Same comments as for fig 9 and 10: it would be better to use a different colormap for figures a and b, and perhaps group all figures together to prevent them spreading over multiple pages. Also reconsider using relative/normalized metrics.

    Indeed. The colour tables used in the figures have been homogenised and we now use the SS statistic to compare to the base simulation. Please see our answer to 23.

28. P20 L6: "at hub height". Does this mean that only ∼100 m was used for all tables? So far I wasn't sure, but I was under the impression that the metrics were calculated on the basis of all measurement heights. What does this mean for the representation of the (distribution of) wind shear between the various model

simulations? I know that the mean profiles have very similar shear, but beware that instantaneous profiles can show substantial variation!

All the statistics have been computed for the levels underlined in Table 1. As the reviewer suggests, the behaviour of the wind statistics could be very different at e.g. 10 meters. We focus on heights relevant for wind energy development. We have added the height of the validation to all figure captions.

For lack of space we concentrate on a single level and do not consider the shear distributions. These have been analysed in a previous paper [4].

29. P20 L7: "Unfortunately, we did not run... so we cannot". This statement contributes substantially to the overall impression that this manuscript is an accomplished fact.

The sentence has been rewritten. However, it is a fact that the project is complete, and while some further simulations would have been very interesting to do, they were not able to be accomplished during the period of this study.

30. P21 L3: It seems a bit weird that this is the last experiment. If I would have designed this experiment, it would have been the first, as the other settings may depend on it. Especially the combination of domain size and nudging/initialization strategy seems influential.

Yes, agreed. This was not the last experiment to be done. It was done at a similar time than the NW domain sensitivities. However, it seemed to fit better here for the flow of the manuscript. In addition, the production run did not follow the advice of using smaller domains. We just hope that the results described here are useful to someone else in the future.

31. P21 L8: This is an interesting dilemma. Did the authors modify the WRF registry to output only the relevant parameters? Would the "restart" option not lift this constraint as the simulation time could be shortened to enable intermediate postprocessing? And how does the pan-European domain compare to the CONUS domain used in the rapid refresh configuration of NOAA? Have the authors contacted them for advise about their reference setup and HPC strategy? Options to stream the WRF output, or to access model fields during a simulation to postprocess them right away would be very welcome recommendations for model development. I wonder whether such features are already available, for example through the "basic model interface" developed by CSDMS.

The creation of the simulations and the postprocessing of the output for a wind atlas is quite different from that of NWP output. For example, the histograms of combined wind speed and wind direction distribution are need for each model grid point and height for the complete duration of the model simulation. There are more details on the post-processing in the companion paper [1].

32. P21 L11: "outside region of interest . . . would be wasted". I have to disagree here. Although it is not the explicit goal of the NEWA project, these data could be very useful for those non-EU countries. Again, please broaden the scope from "NEWA" to "a relevant and interesting dataset for the audience of GMD". I think this dataset can have more impact if it would be available for other researchers as well. The term "waste" therefore rubs me the wrong way.

Agreed. The sentence was a bit harsh and has been rewritten. For context, the computational and time resources were limited — the simulations took 6 months to compute, and every available CPU hour allocated in the PRACE grant for using the cluster at the Barcelona Supercomputer centre (BSC) was used. Also, the data storage is nearly 160 TB, which took almost as long to transfer from the BSC as it did to compute. The scarcity of resources meant that the resources available had to be used frugally. Therefore, while we would have liked to extend the spatial coverage and scientific questions we could answer, hard decisions had to be made to prioritise what could be feasibly handled given the resources available.

33. P21 L26: (and possibly . . . not shown). Model runs that would show this have also not been described as far as I can see. What additional simulations did the authors perform that inspire this statement, or is it mere speculation?

A simulation with a large outer domain and a small inner domain was carried out. The mean wind speed from this extra simulation resembles that of the LG simulation more than the SM simulation. From this, we infer that the position of the inflow boundary is important, but more research is needed. To keep the discussion brief this simulation was not added to the original manuscript. We have rewritten the sentence in the updated manuscript removing the mention of additional simulations that have not been explained.

34. P21 L34: "We decided, however, against very small domains. In terms of accuracy they would probably perform better". Not only does this statement sound speculative, it also partly undermines the objective of the paper. If one of the options considered (the SM domains) was not an option to begin with, why test it? For some sites, the impact of this decision seems to be larger than the accuracy gained through the detailed optimization of all other settings of the model. . .

Yes, agreed. It is contradictory, but we hope that this would be useful information to future wind modellers. Please see our answer to question 30.

35. P22 Fig13: The y-axis is unreadable.

Apologies. The figure has now been replaced with the heatmaps tables to match previous discussion.

36. P24 L2-3: "In that paper we conclude that..." ? Better than just using "raw" ERA5 data?

Agreed. The sentence is too strong and perhaps not relevant here. It has been rewritten.

37. P24 L4: "some questions remain unresolved ... expensive nature of the numerical experiments". This is obviously true, but I feel there are many more questions unanswered because of limited manpower. I'd really appreciate it if the authors could reflect more on that aspect of their study.

    Agreed. Many questions always remain unanswered. And because of the limited set of observations this was not the best region to carry out the many test needed. However, this study still has done much more than previous wind atlas studies. The project has ended and there is limited funding to continue the analysis of the results.

    Some issues regarding the diurnal cycle in the observations and the model simulations are currently being studied. We hope a new publication will result from that analysis.

38. P24 L17: "It would have been optimal..." again this contributes to the "accomplished fact" feeling. This can still be done, can't it? And it can answer some of the questions I have asked, e.g. P5 Fig1 related to the representativeness of the northern domains for Southern Europe.

    Unfortunately this is not possible, due to the limited availability of relevant datasets for validation. However, Part 2 [1] of the study shows very good comparison of the WRF simulations using the final configuration against the Vestas tall tower data. However, because the data is proprietary, the geographical distribution of the errors cannot be shown. Figure 10 of that study shows larger errors in, for example, Turkey compared to the northern sites, but the larger number of sites in France does not show the same tendencies. In addition, we have added to the revised manuscript that "This large dataset can be further verified as additional data becomes available". Hopefully this helps alleviate the "accomplished fact".

    In the revised manuscript we will strengthen the connection between the two parts of the study. This should help clarify many aspects of part 1.

39. P24: The discussion (or other parts of the paper if appropriate) should also address why vertical resolution was not subject to sensitivity analysis, what the uncertainty of the observations is, why wind direction is not considered at all, whether performance is similar across different heights, why/how wind shear has (not) been assessed, how the set-up compares to other similar efforts. The outlook should offer some advice for future studies: what have we learned from this study, in what direction should model development evolve, what are the main strengths/weaknesses of the WRF setup, which parameterisation schemes should we abandon right away, etc.

Agreed. Further discussion has been added. In early experiments, not reported in the manuscript, we experimented with increasing the number of vertical levels from 61 to 91. The results of that experiment showed very small differences between the simulations.

**References**

[1]  Martin Dörenkämper et al. "The Production of the New European Wind Atlas, Part 2: Production and Validation". In: *Geosci. Model Dev. Discuss.* in review (2020).

[2]  Caroline Draxl et al. "Evaluating winds and vertical wind shear from WRF model forecasts using seven PBL schemes". In: *Wind Energy* 17 (2014), pp. 39–55. DOI: `10.1002/we.1555`.

[3]  Filippo Giorgi. "Thirty Years of Regional Climate Modeling: Where Are We and Where Are We Going next?" In: *J. Geophys. Res. Atmos.* (June 2019), 2018JD030094. ISSN: 2169-897X. DOI: `10.1029/2018JD030094`. URL: `https://onlinelibrary.wiley.com/doi/abs/10.1029/2018JD030094`.

[4]  Andrea N Hahmann et al. "Wind climate estimation using WRF model output: Method and model sensitivities over the sea". In: *international Journal of Climatology* 35 (2015), pp. 3422–3439. DOI: `10.1002/joc.4217`.

[5]  Noam Lupu, Lucía Selios, Zach Warner, et al. "A new measure of congruence: The Earth Mover's Distance". In: *Political Analysis* 25.1 (2017), pp. 95–113.

[6]  Julien Rabin, Julie Delon, and Yann Gousseau. "Circular Earth Mover's Distance for the comparison of local features". In: *2008 19th Int. Conf. Pattern Recognit.* IEEE, Dec. 2008, pp. 1–4. ISBN: 978-1-4244-2174-9. DOI: `10.1109/ICPR.2008.4761372`. URL: `http://10.0.4.85/icpr.2008.4761372%20https://dx.doi.org/10.1109/ICPR.2008.4761372%20http://ieeexplore.ieee.org/document/4761372/`.

[7]  Claire L. Vincent and A. N. Hahmann. "The Impact of Grid and Spectral Nudging on the Variance of the Near-Surface Wind Speed". In: *Journal of Applied Meteorology and Climatology* 54 (2015), pp. 1021–1038. DOI: `10.1175/JAMC-D-14-0047.1`.

**Editorial remarks**

1. Excessive use of commas and conjunctions make parts of the text difficult to read. this can easily be addressed by making shorter sentences. For example:

   Yes, we agree. We have revised the manuscript and shortened sentences whenever possible. The specific sentences have been corrected as listed below.

   - P2 L6-7: rewrite "but not only". Very wordy sentence, it has been rewritten.

- P2 L7-8: use only on of "for example ... to name a few". Fixed.
- P2 L13-15: suggest " ... its usefulness. It has ...". Done.
- P3 L13-16: start new sentence at "however". Done.
- P3 L10: start new sentence at "however". Done.
- P3 L12: "A large number" or "Large numbers of" . Done. We have also removed unnecessary words from the sentence.
- P3 L16-18: move "has been reported" to beginning: "a number of studies report ...". Done.
- P3 L19: remove comma after "cases", suggest: "two processes with opposing effects" (remove "canceling each other out") Done.
- P3 L21: citation without brackets. Fixed
- P3 L28: coastal winds? Flow is ambiguous (air or water). We clarified that we refer to the atmospheric flow in the coastal zone.
- P4 L8: Simulations (plural), or perhaps "reference configuration"? Yes, it should be simulations.
- P6 L23: remove "in". Removed.
- P7 L12: suggest: "Small changes in wind speed are (thus) amplified when converted to power." Done.
- P8 L18-20: suggest to split in 2 or 3 shorter sentences. Remove "the aim was", as the next sentence also states "the objective". Done.
- P8 L20: remove "left" (or write "left untouched"?) Done. Agreed. This was a very complex sentence. It was been simplified by using parenthesis.
- P8 L22: "or if there were regional differences" can be omitted as it is already implied by the use of "whether". Done.
- P10 L4: better to split up and rephrase, instead of using "but" twice in the same sentence. Done.
- P10 L6: this sentence can also be split in two shorter sentences.
- P11 L6: Unclear, long sentence. Done.
- P15 L2: "regional" should be "region(s)"? Done.
- P17 L20: "conclusions can be drawn". Done.
- P18 L1: "scheme and run" both refer to a scheme/set-up/configuration, right? Yes, the name "simulations" has been used instead.
- P21 L21: "six" instead of 6 (in line with the surrounding text) Done.
- P21 L34: rephrase "which would face". The two sentences have been rewritten

- P22 L8: unclear sentence; a.g.l. and AGL are the same. Which figures? Indeed. Very confusing sentence. It has been rewritten.

- P23 L18: weird use of commas around ". . . change source...". Done. Should have been "changing the source..."

- P24 L2: "wind climate". Done.

- P24 L6-7: "however... but ...". Done.

- P24 L10: "best optimal" . Missing verb. Fixed.

- P24 L17: "observational dataset". Added.

---

## Author Comment (AC2) · 26 Jun 2020

**Response to Referee #2**

Thank you for the comprehensive comments. Your comments, together with those of referee #1, led to a thorough revision of the paper.

The most general comments regarding the revisions to the manuscript are:

1. At the start of the research project typically there are high expectations placed on the sensitivity experiments, however, reality always brings some corrections and caveats. Given the enormous possibilities in setting up WRF, an "optimal" configuration is unreachable. We have tried to revise the introduction to convey that the paper focuses on finding the "best possible" model configuration **constrained** by the practical issues in running the model simulations and the ultimate goal to use the simulations for a **wind atlas**.

2. The manuscript aims to tell the story of how the NEWA wind atlas came to be. Therefore, further analysis of the model results will make the flow of the paper less clear. We have tried to enhance this structure in the revised manuscript.

3. We have replaced some of the figures (6, 9–12, 13) to homogenise the analysis of the results. We have also added new figures including the RMSE and circular EMD for wind direction.

4. We strengthened the connection to the companion paper, https://www.geosci-model-dev-discuss.net/gmd-2020-23/, which is now available.

The reviewers' comments are in black and our responses in blue.

**General Comments**

1. The paper summarizes an exhaustive sensitivity analysis performed to inform the final model setup of the New European Wind Atlas. This surely must be the most extensive such analysis to date and overall is an impressive achievement. The novel use of the Earth Mover's Distance is also applauded and clearly offers a much-needed complimentary metric alongside the typical timeseries-based performance metrics.

   Thank you. As described above, we have expanded the use of the EMD and Circular EMD (CEMD) for wind direction in the manuscript.

2. I believe this paper should ultimately be published; however, I have several comments and concerns about the work that have not been addressed in the paper. First, all of the critical validation was performed in Northern Europe, despite the NEWA being produced for Europe and Turkey as a whole. I realize that computational expense and data availability/quality were probably a factor, I can't help but feel that with such collaboration across European institutes that

a more regionally diverse validation campaign could have been performed. Of course NEWA has already been produced, but I think some critical commentary on how validation in Northern Europe (with its unique climatology) would apply across other climates in Europe with their own unique climatologies is needed here. Otherwise, the paper reads as if the idea of more extensive validation was overlooked.

In the second part of this study [1], the final wind atlas is validated against masts over all of Europe. However, at the time we did the sensitivity simulations and we needed to decide on a final configuration, further evaluation with data besides the 8 sites in N. Europe was not possible. The public data from tall masts needed for evaluation are scarce over Europe. In a recent paper and database [3], where a global database of tall masts was compiled, there is only a handful of mast over Europe where data are available and only a couple lie in the region chosen for the sensitivity study. Further, even if the data used in part 2 would have been available, the evaluation would not have been possible. There are no masts in Denmark and very few in Germany and the masts in Poland have data for only a few months in 2015. (see Figure 4 of Part 2)

3. Furthermore, I did not find sufficient presentation of results to justify selection of the final model setup. Rather, a wind profile plot and two heat maps of bias and EMD were provided, and it seemed very quickly the section was wrapped up with the final model selection. I think some further synthesis is required, such as a table of figure showing mean bias, RMSE, EMD, etc. across all validation sites. Without this, in my opinion, the selection of the final model setup seems unjustified.

Agreed. The new manuscript includes further figures with all the statistics including BIAS, RMSE, EMD for wind speed and CEMD for the wind direction.

4. Finally, as far as I can tell, ERA-interim was used in the sensitivity analysis, but ERA-5 was used in the final production run. This point is not discussed in this paper but I think it's an important one. Does existing research suggest bias or EMD differences between the two data sets? If so, what are the implications on selecting the best model setup using one large-scale forcing but pivoting to a new product for the actual production runs?

There seems to be some confusion here and we will clarify the data used in the manuscript. The sensitivity simulations described in sections 5.1, 5.2 and 5.5 used ERA-Interim as forcing. All other simulations (excluding the simulation named "ERAI" in section 5.4) used ERA5 data. Actually the sensitivity test of replacing ERA5 with ERA-Interim is described in Table 5 and the results are shown in the heatmap plots. The differences in BIAS and EMD (figures 11 and 12 in the manuscript) show very small differences. Since ERA-Interim was scheduled to be

discontinued in 2019, we chose to continue our sensitivity studies and production run with the ERA5 dataset.

5. In conclusion, I think this is a valuable contribution to the literature. However, several key limitations of this study need to be sufficiently addressed and discussed before final publication. In addition, a couple summary figures and tables would help justify final model selection.

Thank you. We believe the document is much improved after this round of revisions.

**Specific Comments**

1. Page 1, Line 9: Why were sensitivity experiments only conducted in Northern Europe when the data set was for Europe as a whole? Surely tall masts must be available elsewhere? If this was a decision based on computational restrictions, this should be stated and the implications of this smaller validation domain, in the context of regional wind climates, should be discussed.

The sensitivity analysis was done only for the domain over Northern Europe. As mentioned in our answer to item 2 above, tall masts of good quality data publicly available are very scarce. The implications are the focus of section 5.1, where we argue that the behaviour of the mean wind speed relative to various PBL/SL parameterisations is similar among the five domains in very distinct wind climates.

2. Page 2, Line 15: Can 'linearized model' be described more, or at least a couple references listed to provide background?

The paragraph where this statement appears has now been rewritten. We refer here to the whole wind atlas method, which is now described in a little more detail. Therefore, "linearised model" is now "linearised method".

3. Figure 1: As in comment in Line 9, validation only in Northern Europe poses a problem for a product that covers Europe as a whole. This key study limitation needs to be discussed in detail.

This is now better explained in section 5.1. It is a limitation of the study, but we had no alternatives. In retrospect, the validation in paper 2 [1] shows that the resulting wind atlas provides good estimates not only in N. Europe, but also in other regions.

4. Table 1: What is the time resolution of the observed data used to indicate sample size? I'd assume hourly but please make this clear.

We have added a description of the temporal resolution of the data, which was 10-min means that were filtered to hourly using the period closest to the top of

the hour. Additionally, Table 1 has been extended to include wind directions, and show the data availability as a percentage rather than number of samples.

5. Page 6, Line 9: Given the known impact of turbine wakes at these measurement sites, why not filter the data by wind direction to ensure the data are free stream? Especially in such a detailed sensitivity analysis where performance metrics between different model setups can be on the order of 0.1 m/s, allowing wakes to affect the measurement data seems inappropriate.

   We now explain in the text that the impact of the wind farm is difficult to quantify. For example, at some of the sites, wind farms were being built and tested in 2015, and without operational data, we cannot know when a wind farm was curtailed or otherwise not operating. Additionally, filtering for the wind farm possible perturbation can severely decrease the number of samples for some sites. We have now added the centre of the filtering wind direction and the fact that there is an additional wind farm near the FINO2 mast. The text in the revised manuscript has been changed to "...the data has not been filtered or corrected for the turbine wakes. However, the presence of the wind farm can impact the evaluation of the model results and should be kept in mind."

6. Page 7, Line 14: I'd use 'interpreted' rather than 'understood' when describing EMD as a measure of physical work.

   Agreed. We have replaced "understood" by "interpreted"

7. Page 7, Line 15: Given the novelty of the EMD metric, I wonder if a new Figure showing the area between cumulative distribution functions would be useful, given this is how the metric is actually computed.

   Agreed. This is a very good suggestion. A new panel has been added to Figure 2 showing the cumulative distribution functions.

8. Page 7, Line 16: What are circular variables and why are they relevant here? Are you validating wind direction?

   Circular variables are variables, like wind direction, where there is an apparent discontinuity at between 0 and 360°. In the updated manuscript we use a version of EMD metric adapted for circular variables (CEMD), which was used to evaluate the wind direction distributions in the model simulations.

9. Page 8, Line 23: Why was WRF 3.6.1 used, given it is 6 years old and the significant advances made since then? Was this part of an older study that is now being published?

   At the start of the project (summer 2015), WRF V3.6.1 was not that old (it was released August 14, 2014) and it provided good evaluation against observations in other regions, for example South Africa [2]. Using the latest version of a model is not always advantageous as seen in the changes to the MYNN parameterisation

in WRF V3.8.1, that heavily impacted the validation statistics. Later in the large ensemble we moved to WRF V3.8.1. We acknowledge that the model version used in the various simulations was not clearly stated. This situation is fixed in the revised manuscript.

10. Page 10, Line 3: But MYNN winds are higher in the NW offshore domain and lower in the SW domain. Can you discuss? Is NW offshore domain generally more stable?

What was meant by the statement was that normally the winds in the YSU scheme are larger than those in the MYNN scheme (see Figure 1 in the answer to the comments from reviewer #2). But when conditions are mostly unstable, as it is in the French Atlantic coast (50–60% of the time), Mediterranean sea (60–70% of the time) and some coastal areas Turkey, the situation reverses and the 100m mean winds are higher in the simulations using the MYNN scheme than the YSU scheme. Yes, conditions are mostly stable or neutral over the North Sea and the Baltic Sea. The sentence in the text has been expanded to clarify this issue.

11. Figure 6: Given the detailed justification of EMD earlier, why is it not being used here?

Totally agree. At the time that these analyses were originally made we had yet to discover the advantages of the EMD metric. But now we show this metric throughout the manuscript and also in Figure 6.

12. Figure 8: I'm struggling trying to distinguish the different model runs. Multiple setups seem to have identical markers (at least to the naked eye). Also the lines are so tightly clustered that it's generally not possible to discern one profile from another. As such the Figure does not provide much useful information and I would recommend revising or deleting.

The objective of the figure was to show that the wind profiles from the simulations clustered, not to be able to differentiate between them. We have redone the figure with a single grey colour, highlighting only the results from two relevant simulations. Hopefully it will reflect better our intention.

13. Figure 9a: Would an additional column showing average across sites be useful in identifying the best performing model setup?

Thanks. It is a good suggestion. However, in this case we would argue against adding the averaging over the stations. Please see the long discussion in the response to Referee #1, item 25 (P18 L5). We don't want to give the impression that the decision on final configuration was just based on a raw evaluation of the numbers. Adding the average over the stations will convey that impression in our opinion.

14. Figure 9b: I'm not sure I see the value of performance metrics relative to the 'base' setup. In my mind this base setup is just another member of the ensemble

and not otherwise special. So why compare all ensembles against this one? Do we know it to be the most accurate? If not, I don't see the value in this relative comparison. Please justify.

Thank you, this is an important question. As we searched for the "best" model configuration, we kept asking "Is there another different configuration that will be better than our base?" The relative heatmaps help answer that, while also showing how small the differences between simulations are, which is sometimes hard to spot in the BIAS or EMD plots alone. This is because the differences between the stations are often more pronounced, for absolute values of metrics, than the differences between the ensemble members at the same station. It is important to note, that this method of examining results does not assume that the "BASE" setup is the most accurate, it is just a more convenient way of identifying differences between different models.

15. Figure 10b: Likewise to comment above. I'm not seeing the value of this relative comparison.

Please see our reasoning above.

16. Page 18, Line 5: This is a big jump to conclude the best performing model setup based on the figures shown in this section. For example, the improved performance of MO over the Base and MM5 setups isn't clear from the profile plots or the heat maps. I think some final figure or table is needed showing key performance metrics averaged across all sites in order to justify this model choice. It also seems that the multi-physics sensitivity analyses and the selection of final production run in Section 5.3 was done using ERA-interim as the large scale forcing in WRF. However, ERA-5 was used in the final NEWA. This seems problematic given potential differences (e.g., biases) between the two data sets. I understand that ERA-5 was not available at the time these simulations were performed; however, some discussion around the implications of changing the large scale forcing without sensitivity analysis needs to be provided.

Agreed. It is a very fair question. In conclusion the simulations show that many parameters usually though to be important for NWP or climate modelling have little or no influence. So it would probably be fair to choose any of them, except for some PBL/LS/LSM combinations that definitely degrade the validation metrics. A long discussion on the matter was given in the answer to referee #1 (item 25, P18, L5). In the revised manuscript we try to convey this in a more direct manner.

17. Page 19, Line 8: Unclear how ERA5 reanalysis slow down of winds relates to a sensitivity analysis of ERA-interim, FNL, and MERRA2. Was ERA5 part of this comparison?

All experiments in these tables used ERA5. The comparison is then from ERA5 to MERRA2, FNL, and ERAI. As the table reveals the differences between ERAI

and ERA5 are small.

18. Figure 11a and 12a: What is the difference between BASE and ERAI? I thought the base run was done using ERA-interim.

   All the experiments, except for "ERAI", were carried out using ERA5.

19. Figure 11b and 12b: Same comment as previous.

   Same response as above. Sorry about the confusion.

**References**

[1]  Martin Dörenkämper et al. "The Production of the New European Wind Atlas, Part 2: Production and Validation". In: *Geosci. Model Dev. Discuss.* in review (2020).

[2]  Andrea N. Hahmann et al. *Mesoscale modeling for the wind atlas for South Africa (WASA) Project.* Tech. rep. DTU Wind Energy, p. 77. URL: `http://orbit.dtu.dk/services/downloadRegister/107110172/DTU_Wind_Energy_E_0050.pdf`.

[3]  J. Ramon et al. "The Tall Tower Dataset: a unique initiative to boost wind energy research". In: *Earth System Science Data* 12.1 (2020), pp. 429–439. DOI: `10.5194/essd-12-429-2020`. URL: `https://essd.copernicus.org/articles/12/429/2020/`.

---

## Author Response (AR2)

**Response to Referee comments — Second review**

August 27, 2020

Thank you once again for very helpful comments.
The reviewers' comments are in black and our responses in blue.

The manuscript has improved substantially from the initial submission, and I would be more than happy to see it published. Especially the extensive tables with scores for each configuration are an excellent resource for future reference. I'm also happy with the authors' commitment to making (an excerpt of) the data publicly available. Thank you. The summary statistics from nearly all (a single run is missing due to corrupted data) simulations are now available from Zenodo, DOI: 10.5281/zenodo.4002351.

1. I just have a few small remarks that the authors might still want to address:

   - P4 L22 "not warranted" I would suggest to phrase this is bit more carefully. Agreed. The new sentence reads "Alternatively, a single continuous run can be performed. Although the continuous simulation may present certain advantages, like preserving the memory of land-atmosphere processes (Jimenez et al 2011), it is not necessarily more accurate. Additionally, this type of simulation increases the wall clock time required to complete the runs by at least an order of magnitude and thus make long-term high-resolution simulations not viable."

   - P12 L12: It would be good to end this section with a concluding sentence (e.g. move/use P13 L3). Agreed. Two sentences have been added before and after the start of the new section. "Summarising, the effect on the wind speed when one of those PBL schemes is replaced with another is nearly insensitive to the location of the domain (i.e., North versus South)" before and "We evaluate the mean wind profiles from the four model setups in Sec. 5.1 to observations in the NW domain to proceed in determining a suitable model configuration. " at the start of the next section.

   - P14 L4: It would be clearer to start this section with something like: "Having set our baseline configuration, we now start...". Currently you write "we start" while the reader feels like we've started 3 pages ago. Agreed. It has been changed to "Having set our baseline configuration, we now start testing other possible modifications to the model setup and parameterisation options. "

   - Colormaps: the colormaps are better than before, but sometimes the use of a diverging colormap for a continuous model score is still slightly confusing.

For a diverging score like bias, white is best, while for a continuous score like RMSE, purple is best. I think I would only use the brown part of the colormap here. We suggest a compromise here. The colour table for the biases is now brown for large positive and negative values and purple for values close to zero.

2. And some technical corrections:

   - P2 L20 "in two parts"? Done

   - P4 L21 "type of continuous ... may" Done

   - P4 L22 "its is not". Done

   - P9 L23 "the ERA-Interim ..." ? Done

   - P18 L17 difficult to read/wrong use of commas. Done

   - P19 L24 "similar than" Done

[revised manuscript text omitted]